# ARGOS: Hierarchical Autoregressive Generation of Unbounded 3D Outdoor Scenes with High Fidelity and Spatial Control

## Abstract

We present ARGOS, a hierarchical autoregressive framework for generating unbounded 3D outdoor scenes with high fidelity and spatial control. Existing methods for large-scale 3D scene generation are limited by a fundamental trade-off between global consistency and fine geometric detail. While prior diffusion-based approaches struggle with long-range coherence, our framework resolves this tension by decoupling the challenge into two stages. First, a causal autoregressive model establishes a globally coherent layout by processing scene chunks in sequence. Second, a masked autoregressive model generates detailed local geometry conditioned on this global layout and its neighbors. These geometric latents are then decoded by our enhanced VAE to ensure high-fidelity reconstruction. To enable user control, we introduce an automated pipeline that extracts complex spatial relationships from scenes, producing a structured dataset that allows ARGOS to follow precise text-based commands. Comprehensive experiments demonstrate that ARGOS significantly outperforms existing methods in unconditional generation, achieving superior FPD and KPD metrics across multiple scales. For text-conditioned synthesis, our approach excels at generating coherent, large-scale scenes that precisely adhere to complex spatial instructions.

## 1 Introduction

The synthesis of large-scale, detailed, and coherent 3D scenes represents a core challenge in computer graphics and vision, with significant implications for gaming, simulation, and world models. The core difficulty lies in a fundamental tension: achieving global consistency across vast environments while preserving the intricate, localized geometric details necessary for immersion and interaction. This demands models capable of joint reasoning across macro-level spatial layouts and micro-level surface geometry, often over unbounded domains.

Current approaches to 3D scene generation largely follow two paradigms. The first lifts 2D diffusion priors (Rombach et al., 2022; Blattmann et al., 2023) into 3D via multi-view reconstruction (Chung et al., 2023; Liang et al., 2025; Yu et al., 2025). While strong on global appearance, this approach often yields geometries with artifacts and inconsistencies. The second paradigm directly generates explicit 3D mesh geometry, typically by using native 3D diffusion models to sequentially synthesize volumetric chunks (Lee et al., 2025; Wu et al., 2024b; Ren et al., 2024a; Meng et al., 2025).

While more direct, this paradigm faces three significant challenges. First, while these methods can generate high-quality bounded shapes, they fail to achieve long-range coherence in unbounded scenes. This failure is fundamental to their core operational principle: a reliance on a fixed-context, iterative denoising process that is inherently local. Consequently, scene extension strategies like resampling-based inpainting (Lugmayr et al., 2022) and chunk-based outpainting (Lee et al., 2025) inevitably produce visible seams and logical inconsistencies at chunk boundaries. Second, its underlying VAEs often produce low-fidelity reconstructions. Third, it provides no robust mechanisms for controllable synthesis from high-level commands.

In contrast to the local nature of diffusion, the autoregressive (AR) paradigm excels at modeling long-range dependencies, as evidenced by its success in language modeling (Hurst et al., 2024; Comanici et al., 2025) and visual generation (Tian et al., 2024; Siddiqui et al., 2024). This makes

AR models a natural fit for ensuring global coherence and enabling precise, text-conditioned control. However, a purely causal AR approach, while powerful for global structure, struggles to synthesize fine-grained geometric details. Achieving high-fidelity large-scene 3D generation with the vanilla AR paradigm poses a critical challenge, which is demonstrated by our ablation of various generation frameworks in Sec. 4.4.2.

To this end, we introduce ARGOS, a hierarchical framework designed to harness the global modeling strengths of autoregression while ensuring local geometric fidelity. The core of ARGOS is a hierarchical AR generator that decouples global planning from local synthesis: a causal AR model first establishes a coherent global layout, upon which a masked bidirectional AR model then synthesizes high-fidelity local details. This design directly resolves the fundamental tension between large-scale structure and fine-grained geometry. Further, the AR generator is built upon an enhanced 3D chunk-based VAE that employs direct mesh partitioning for occupancy supervision at higher resolution, augmented by salient geometry guidance derived from sharp-edge sampled points (Chen et al., 2025) and surface normals, thereby establishing a robust foundation for high-quality geometric reconstruction. Finally, to enable precise interactive control, we develop an automated annotation pipeline that produces a structured text dataset for training high-level spatial conditioning. We validate our approach on large-scale unbounded 3D scene generation. ARGOS demonstrates superior performance in both unconditional synthesis (measured by FPD/KPD) and text-conditioned generation, excelling in quality, scale, and controllability.

## 2 RELATED WORK

### 2.1 3D SCENE GENERATION METHODS

The field of 3D scene generation has evolved along two primary paradigms based on their geometric representations. The first employs 2D-to-3D lifting pipelines that leverage strong image generation priors. Early works pioneered perpetual view generation from single images (Liu et al., 2021; Li et al., 2022), which later extended to text-conditioned radiance field generation Chen et al. (2023); Höllein et al. (2023). More recent approaches use 2D diffusion models to generate multi-view content before lifting it to 3D (Yu et al., 2024; 2025; Chung et al., 2023; Li et al., 2024a; Zhou et al., 2024). While these methods create diverse scenes, they suffer from a fundamental limitation: geometric quality is compromised by accumulated depth prediction errors and view synthesis inconsistencies. These geometric shortcomings motivate the second paradigm: direct 3D generation. Within this paradigm, diffusion-based methods like BlockFusion (Wu et al., 2024b), NuiScene (Lee et al., 2025), and LT3SD (Meng et al., 2025) synthesize scenes by learning to denoise structured latent representations such as triplanes or vector sets. While these methods achieve better geometric quality than 2D-to-3D approaches, they remain predominantly diffusion-based and struggle with unbounded generation due to their reliance on local context modeling. These limitations are addressed by our hybrid autoregressive framework, which decouples global layout planning from local detail synthesis to handle large-scale environments.

Current methods for controllable 3D generation often rely on structured priors that pose significant accessibility challenges for average users. One group relies on complex structural priors, such as the scene graphs required by GraphDreamer (Gao et al., 2024), CommonScenes (Zhai et al., 2023), InstructLayout (Lin et al., 2025) and C3DO-SG (Liu et al., 2025), or the geographic data (like 2.5D OSM data for UrbanWorld (Shang et al., 2024) and HD maps for InfiniCube (Liu et al., 2021)) which are difficult to obtain. Another group, including Holodeck (Yang et al., 2024) and SceneCraft (Hu et al., 2024), utilizes large language models for spatial arrangement but is limited to manipulating predefined asset libraries and cannot generate new geometry. SSEditor (Zheng & Liang, 2024) offers a mask-conditional approach for semantic scene generation and editing using a diffusion model, providing a different form of control. Even text-driven methods like Text2LiDAR (Wu et al., 2024a) remain constrained by their output modality, generating sparse point clouds where text primarily controls semantic content rather than detailed spatial layout. This widespread reliance on hard-to-acquire priors or limited control modalities highlights a critical gap in achieving accessible, fine-grained control over both the generation and spatial arrangement of novel 3D geometry. Our work addresses this by introducing a purely text-driven system that generates complete 3D meshes from natural language.

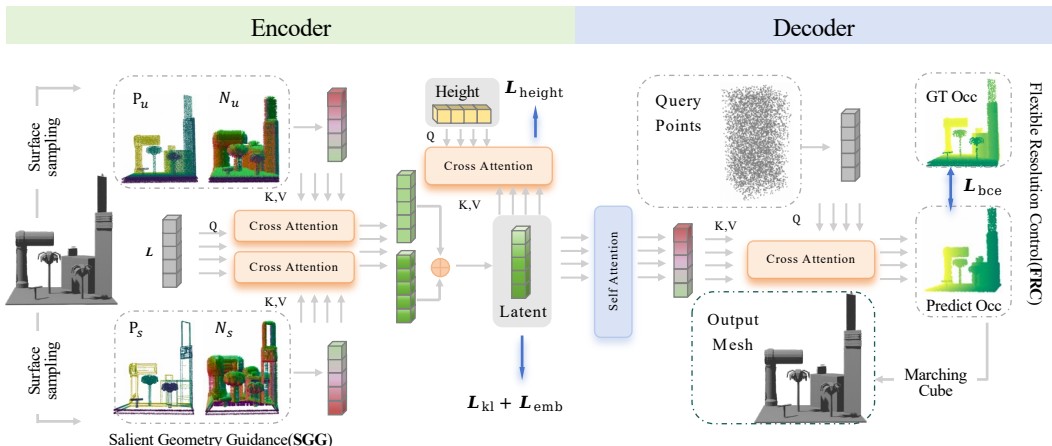

Figure 1: **Illustration of ARGOS 3D chunk VAE**. ARGOS VAE introduces high resolution supervision sand salient geometry guidance to improve the reconstruction fidelity.

## 2.2 LARGE-SCALE UNBOUNDED SCENE GENERATION

Generating unbounded 3D scenes while maintaining spatial coherence presents a significant challenge. Most diffusion-based methods divide scenes into equal chunks and adopt chunk-based generation with local context modeling (Wu et al., 2024b; Lee et al., 2024; Liu et al., 2024). These fixed-context approaches commonly use overlapping windows and resampling-based inpainting techniques like RePaint (Lugmayr et al., 2022) for scene extension. However, this strategy introduces artifacts at chunk boundaries due to insufficient global context propagation. While explicit outpainting models (Lee et al., 2025) can improve boundary coherence and inference speed, achieving truly seamless spatial transitions across large-scale environments remains an open problem. In contrast, our autoregressive framework models dependencies between chunks sequentially, ensuring global coherence by design rather than as a post-hoc repair strategy like inpainting.

## 3 METHODOLOGY

### 3.1 VAE FOR HIGH FIDELITY CHUNK GEOMETRY

ARGOS AVE is build upon the chunk VAE from NuiScene (Lee et al., 2025) (see A.2 for the preliminary). We propose an enhanced mesh chunk VAE featuring two key techniques, Flexible Resolution Control (FRC) and Salient Geometry Guidance (SGG), to improve the reconstructed geometry details as shown in Fig. 1.

**Flexible Resolution Control.** The most straightforward approach to enhance chunk geometric details is to increase the resolution of the occupancy supervision. In NuiScene (Lee et al., 2025), the data pipeline for generating chunk occupancy proceeds as follows: the scene mesh is first converted into a Signed Distance Field (SDF), which is then thresholded to obtain occupancy values. Chunk occupancy is subsequently sampled using coordinates along the length (L) and width (W) dimensions. However, due to the technical limitation of the mesh2sdf tool (Hu et al., 2019), which only supports up to 32-bit representation, the maximum achievable resolution is constrained to approximately 1k voxels along each axis. Since the current scene resolution already approaches this upper limit, directly increasing the chunk resolution in the above data process pipeline is non-trivial.

To address the limitation, we propose a novel pipeline that directly samples the chunk mesh from the scene mesh. This chunk mesh is then converted into a chunk SDF, from which the high-resolution chunk occupancy is derived and more detailes can be found in Fig.9. This approach elevates the upper resolution limit of chunk occupancy from 50 to 1000 voxels per axis, thereby enabling more flexible and precise control over the chunk occupancy resolution to improve the reconstruction fidelity.

**Salient Geometry Guidance.** In addition to increasing the resolution, we also enhance geometric reconstruction details from the perspective of input points. We define the sharp-edge sampled points proposed in Dora (Chen et al., 2025) and point normal as the salient geometry guidance. Instead

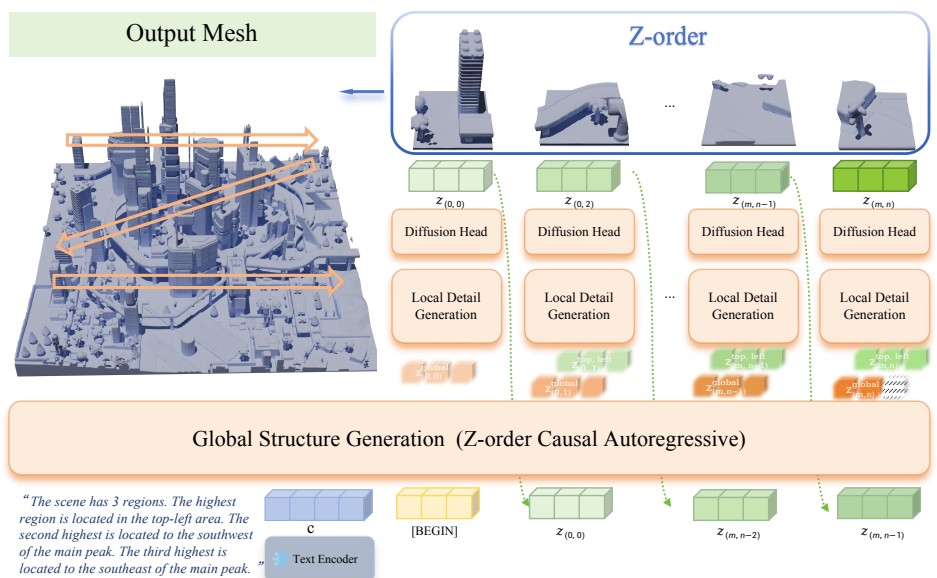

Figure 2: Illustration of hierarchical autoregressive framework and the inference process.

of only using uniformly sampled points, we further incorparate salient geometry guidance for VAE input feature to enhance geometry details. After that, we obtain the latent vector set $Z_u$ and $Z_s$ from uniform points and sharp-edge points by updating the VAE encoder as follows:

$$Z_u = \text{CrossAttn}(L, \text{CAT}(\text{PosEmb}(P_u), \text{PosEmb}(N_u))), \tag{1}$$

$$Z_s = \text{CrossAttn}(L, \text{CAT}(\text{PosEmb}(P_s), \text{PosEmb}(N_s))), \tag{2}$$

where $L$ is the learnable query set, $P_u$ and $P_s$ are the uniform points and sharp-edge points sampled from the scene mesh, $N_u$ and $N_s$ are the corresponding normal maps. The final latent set $Z$ is formed by aggregating features from both sampling strategies via element-wise summation $Z = Z_u + Z_s$.

**Objective optimization.** As for VAE decoder and training loss, we largely follow the formulation from NuiScene as described in the preliminary section except that , enabled by our FRC pipeline, the query points and occupancy supervision can be sampled at a much higher resolution. Chunk VAE optimizes the binary cross entropy loss $\mathcal{L}_{bce}=\text{BCE}(O,\tilde{O})$ between occupancy prediction and ground truth, a KL divergence regularization $\mathcal{L}_{kl}$ of $Z$, a embedding constraint $\mathcal{L}_{emb}=\text{MSE}(Z,\hat{Z})$ and height prediction loss $\mathcal{L}_{height}=\text{MSE}(h,\tilde{h})$. The total optimization loss is formulated as:

$$\mathcal{L}_{\text{VAE}} = \lambda_{bce}\mathcal{L}_{bce} + \lambda_{kl}\mathcal{L}_{kl} + \lambda_{emb}\mathcal{L}_{emb} + \lambda_{height}\mathcal{L}_{height}. \tag{3}$$

### 3.2 HIERARCHICAL AUTOREGRESSIVE FRAMEWORK FOR HIERARCHICAL SCENE SYNTHESIS

Our hierarchical autoregressive framework addresses large-scale 3D scene generation through a unified hierarchical architecture that seamlessly integrates global layout planning and local detail synthesis, enabling both unconditional large-scale generation and optional text-conditioned controllable synthesis.

#### 3.2.1 HIERARCHICAL ARCHITECTURE DESIGN

We model a scene as a grid of $H \times W$ latent chunks, where each chunk $\mathbf{z}_i \in \mathbb{R}^{l \times d}$ is a set of $l$ tokens encoded by VAE. Our framework generates this grid autoregressively, using a hierarchical approach (Fig. 2) that decomposes the task into two stages to balance global coherence and local detail.

**Global Structure Generation.** In the global stage, we treat each chunk as a unit and generate the scene structure following a Z-order spatial traversal pattern. The Z-order traversal naturally captures 2D spatial dependencies while maintaining strict causal ordering. We employ block-wise

causal attention masking that allows the model to access all $l$ tokens within any previously generated chunk. For chunk $i$ at spatial coordinates $(x, y)$, the global autoregressive model predicts:

$$p(\mathbf{z}_i^{\text{global}}|\mathbf{z}_{<i}^{\text{global}}, [\text{BEGIN}], \mathbf{c}) = f_{\text{global}}(\mathbf{z}_{<i}^{\text{global}}, [\text{BEGIN}], \mathbf{c}, \text{RoPE}(x, y)), \tag{4}$$

where $\mathbf{z}_{<i}^{\text{global}}$ represents all previously generated chunk embeddings, $[\text{BEGIN}]$ serves as a special start token for sequence initialization, $\mathbf{c}$ is the conditioning vector (text embedding or $\emptyset$ for unconditional generation), and $\text{RoPE}(x, y)$ encodes 2D spatial coordinates using Rotary Positional Embeddings, which naturally capture geometric relationships between chunks across scene scales.

**Local Detail Generation.** Given $\mathbf{z}_i^{\text{global}}$, the local stage generates the internal structure in chunk $i$. Since point cloud data lacks inherent sequential ordering, we employ masked autoregressive generation with bidirectional attention (Li et al., 2024b), which choice reflects the spatial nature of 3D geometry where relationships are fundamentally non-sequential.

For chunk $i$, we randomly select a masking subset $\mathcal{M}_i \subset \{1, 2, \dots, l\}$ and predict masked tokens autoregressively within the set:

$$p(\mathbf{z}_i^{\mathcal{M}_i}|\mathbf{z}_i^{\mathcal{U}_i}, \mathbf{c}_i^{\text{hierarchical}}) = \prod_{j \in \mathcal{M}_i} p(\mathbf{z}_i^j|\mathbf{z}_i^{\mathcal{U}_i}, \mathbf{z}_i^{k<j}, \mathbf{c}_i^{\text{hierarchical}}), \tag{5}$$

where $\mathcal{U}_i = \{1, 2, \dots, l\} \setminus \mathcal{M}_i$ are unmasked indices, $\mathbf{z}_i^{k<j}$ denotes previously predicted masked tokens with $k \in \mathcal{M}_i, k < j$, and:

$$\mathbf{c}_i^{\text{hierarchical}} = \text{Concat}([\mathbf{z}_{\text{top}}, \mathbf{z}_{\text{left}}, \mathbf{z}_i^{\text{global}}]), \tag{6}$$

where $\mathbf{z}_{\text{top}}$ and $\mathbf{z}_{\text{left}}$ are tokens from the top and left neighboring chunks of chunk $i$, respectively. This hierarchical context provides: (1) structural guidance through $\mathbf{z}_i^{\text{global}}$, and (2) spatial continuity through neighboring chunks, ensuring geometric coherence across chunk boundaries.

### 3.2.2 TRAINING OBJECTIVES AND OPTIMIZATION

Our training combines three complementary objectives that address different aspects of large-scale 3D generation:

$$\mathcal{L}_{\text{total}} = \mathcal{L}_{\text{diffusion}} + \lambda_{\text{coh}}\mathcal{L}_{\text{coherency}} + \lambda_{\text{dpo}}\mathcal{L}_{\text{DPO}}. \tag{7}$$

**Diffusion-Enhanced Token Generation.** To generate high-quality continuous embeddings while avoiding regression-induced blurring, we integrate denoising diffusion within our masked autoregressive framework. For the $j$-th latent token in chunk $i$, the diffusion objective is:

$$\mathcal{L}_{\text{diffusion}} = \mathbb{E}_{i,j,t\sim[1,T],\epsilon\sim\mathcal{N}(0,\mathbf{I})}\left[\left\|\epsilon - \epsilon_\theta(\mathbf{z}_{i,t}^j, t, \mathbf{c}_i^{\text{hierarchical}})\right\|_2^2\right], \tag{8}$$

where $\mathbf{z}_{i,t}^j$ represents the noisy $j$-th token embedding in chunk $i$ at diffusion timestep $t$, and $\epsilon_\theta$ is the learned denoising network. During inference, tokens are initialized with Gaussian noise and iteratively denoised over $T$ steps.

**Coherency-Aware Regularization.** To prevent mode collapse into repetitive patterns, a common failure in large-scale synthesis, we introduce neighborhood similarity regularization that encourages spatial diversity while preserving local coherence:

$$\mathcal{L}_{\text{coherency}} = \mathbb{E}_{i\sim\text{chunks}}\left[1 - \cos(\mathbf{z}_i^{\text{global}}, \mathbf{z}_{\text{neighbor}(i)}^{\text{global}})\right], \tag{9}$$

where $\text{neighbor}(i)$ denotes spatially adjacent chunks (top and left). This cosine-based loss directly maximizes embedding diversity between neighboring chunks, promoting varied yet coherent spatial arrangements.

**Direct Preference Optimization.** For unconditional generation, which is prone to producing empty or repetitive scenes, we employ DPO to learn from human preferences over scene quality:

$$\mathcal{L}_{\text{DPO}} = -\mathbb{E}_{(\mathbf{c},\mathbf{Z}_w,\mathbf{Z}_l)\sim\mathcal{D}}\left[\log\sigma\left(\beta\log\frac{\pi_\theta(\mathbf{Z}_w|\mathbf{c})}{\pi_{\text{ref}}(\mathbf{Z}_w|\mathbf{c})} - \beta\log\frac{\pi_\theta(\mathbf{Z}_l|\mathbf{c})}{\pi_{\text{ref}}(\mathbf{Z}_l|\mathbf{c})}\right)\right], \tag{10}$$

where $\mathbf{c}$ is the conditioning vector (text embedding or $\emptyset$ for unconditional generation), $(\mathbf{Z}_w, \mathbf{Z}_l)$ are preferred and dispreferred scene generation outputs, $\pi_{\mathrm{ref}}$ is the reference model, and $\mathcal{D}$ is our curated preference dataset. DPO effectively addresses mode collapse and repetitive patterns common in large-scale 3D generation, promoting spatially coherent and semantically meaningful scenes.

**Training Details.** We set $\lambda_{\mathrm{coh}} = 0.1$, $\lambda_{\mathrm{dpo}} = 0.05$, $\beta = 0.1$ for DPO. Detailed implementation details are provided in A.5.2.

### 3.2.3 TEXT-CONDITIONED CONTROLLABLE GENERATION

To extend beyond unconditional generation, we incorporate optional text conditioning as a controllability enhancement. This mechanism enables precise scene control through natural language while preserving the framework's capacity for purely unconditional synthesis.

**Automated Scene Description Pipeline.** To enable text conditioning, we develop an automated annotation pipeline converting 3D geometry to natural language descriptions through four stages: (1) extracting 2D height maps from 3D geometry, (2) watershed segmentation for object isolation, (3) computing geometric properties and spatial relationships, and (4) LLM-based caption synthesis from structured geometric data.

**Conditioning Integration and Guidance** Text conditioning is seamlessly integrated into our hierarchical framework. Input prompts are encoded using a frozen CLIP text encoder to produce conditioning vectors $\mathbf{c}$. This vector guides generation at the global level (Eq. 4) and propagates through the hierarchical context (Eq. 6). To enable classifier-free guidance, we randomly drop text conditions during training with probability $p_{\mathrm{drop}} = 0.1$.

## 4 EXPERIMENTS

### 4.1 DATASET COLLECTION

**Data Sources and Preprocessing.** We construct our dataset using 13 large-scale scenes with dense urban structures for fair comparison with NuiScene (Lee et al., 2025). Raw meshes undergo preprocessing to remove artifacts, normalize to $[-1, 1]$ range, scaled according to architectural anchors for consistency, and regularize ground planes through occupancy-based detection and mesh completion.

**Chunk Extraction.** We convert preprocessed meshes to signed distance fields and extract chunks using spatial filtering to exclude regions with ground holes or excessive flatness. Qualified regions are segmented into $0.1 \times 0.1$ grid meshes using Blender's bisect plane operation. Each chunk yields SDF, occupancy grids, point clouds, and normals at higher resolution than NuiScene. We construct datasets with grid sizes of $2 \times 2$, $4 \times 4$, $8 \times 8$, $16 \times 16$, and $8 \times 20$ for both single-scene and 13-scene configurations as detailed in Tab. 1. More data collection details can be found in A.1.

Table 1: Statistics of the collected chunk data of both one scene and 13 scene across multi scale resolutions.Chunk diversity boosts generalization, and multi-resolution support ensures consistent multi-scale spatial generation.

| Dataset | 2×2 | 4×4 | 8×8 | 16×16 | 8×20 |
|---|---|---|---|---|---|
| 1 scene | 100k | 100k | 30k | 4k | 2k |
| 13 scenes | 280k | 200k | 12k | 5k | 4k |

### 4.2 VAE RECONTRUCTION EXPERIMENT

Table 2: Comparisons between Nuiscene VAE and ARGOS VAE across 1 scene and 13 scenes.

| Dataset | Method | IoU (↑) | CD-$p$ (↓) | CD-$n$ (↓) | F-Score (↑) |
|---|---|---|---|---|---|
| 1 scene | Nuiscene VAE (Lee et al., 2025) | 0.4690 | 0.0062 | 0.0212 | 0.8314 |
| | ARGOS VAE (**Ours**) | **0.6686** | **0.0009** | **0.0087** | **0.9696** |
| 13 scenes | Nuiscene VAE (Lee et al., 2025) | 0.4066 | 0.0071 | 0.0223 | 0.7923 |
| | ARGOS VAE (**Ours**) | **0.6114** | **0.0013** | **0.0094** | **0.9567** |

### 4.2.1 EXPERIMENT SETUP

**Dataset.** We evaluate ARGOS-VAE through a pilot study on a single-scene dataset with 2×2 grid configuration, followed by an extended assessment of its generalization capability using a 13-scene dataset under the same grid setting. Each quad-chunk was subdivided into four sub-chunks for VAE training, yielding 400k and 1,120k training samples respectively. From each dataset, 1k samples were held out for quantitative evaluation.

**Baselines.** We adapt the NuiScene VAE with vector set as our baseline. To overcome the reconstruction limitations imposed by fixed occupancy resolution in the original pipeline, we introduce a revised approach: instead of splitting chunks directly from scene-level occupancy, we first extract mesh chunks from the scene mesh and then convert them to occupancy chunks at a flexible resolution. This allows increasing the occupancy resolution from $(50, h_{vox}, 50)$ to $(256, h_{vox}, 256)$, significantly improving the quality of reconstruction. We train the NuiScene VAE and our ARGOS VAE in the new collected datasets and more implementation details can be found in A.5.1.

**Evaluation protocol.** We evaluated the quality of the reconstruction using a comprehensive set of metrics: volumetric IoU (Siddiqui et al., 2021), point cloud Chamfer distance ($CD_p$), normal Chamfer Distance ($CD_n$), and F-score. All metrics are computed between the reconstructed mesh and ground truth using 10k sampled points and normals. Before evaluation, both meshes are normalized to the range $[-1, 1]$. The voxel size for IoU and the threshold for F-score are set to $0.05m$.

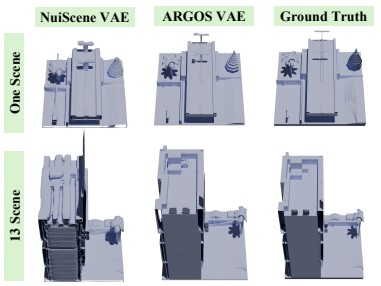

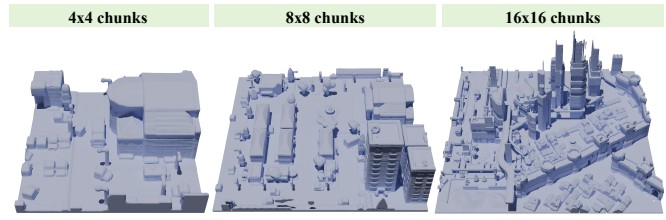

Figure 4: **Example of Spatial Control Generation.** The prompt is: "The scene has 3 regions. The highest region located in the top-right area. The second highest region is located to the west of the main peak. The third highest region is located to the southeast of the main peak."

Figure 3: Qualitative comparisons on reconstructed geometry. Zoom-in can see more details.

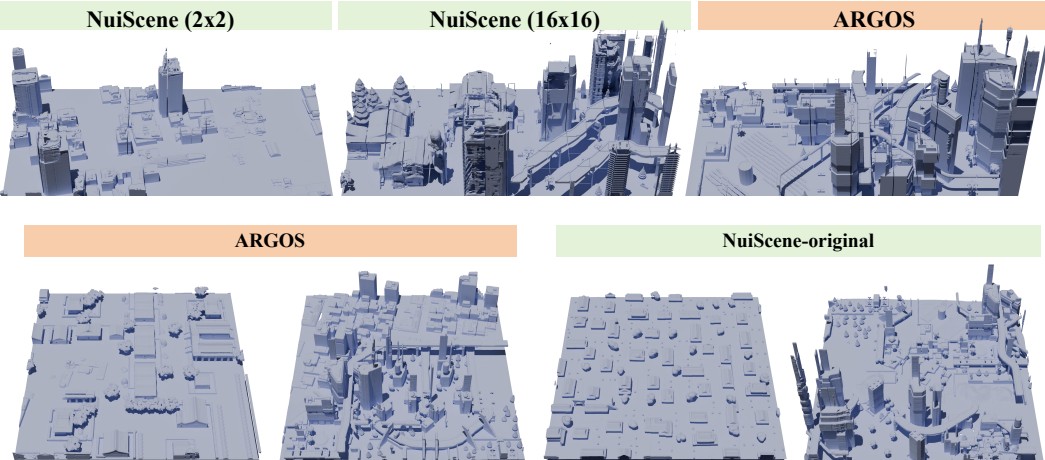

Figure 6: Qualitative comparison using frozen NuiScene VAE on 13-scene data. Even with identical VAE quality, ARGOS generates more coherent layouts, fewer repetitive patterns, and better global consistency compared to NuiScene.

### 4.2.2 RECONSTRUCTION RESULTS

We evaluate the ARGOS VAE from both quantity and quality results across 1 scene and 13 scene dataset. As Tab. 2 shows, ARGOS VAE consistently achieves superior performance than NuiScene

VAE from IoU, $CD_p$, $CD_n$ and F-score metrics. These results demonstrate the effectiveness of our data collection pipeline and VAE architecture. We also render the reconstructed mesh to check the actual visualization improvement. As shown in Fig. 3, compared with the ground truth mesh, ARGOS VAE is capable of reconstructing more accurate geometric details than NuiScene VAE.

### 4.3 AR GENERATION EXPERIMENT

#### 4.3.1 EXPERIMENT SETUP

**Dataset.** We conduct experiments on the dataset of 13 large-scale scenes, partitioned into training and testing sets. For unconditional generation, we evaluate geometric fidelity on both a single scene and the full 13-scene set, using high-resolution chunks $(256, h_{vox}, 256)$. For conditional generation, we assess spatial reasoning and instruction following on the full dataset using lower-resolution chunks $(50, h_{vox}, 50)$ to facilitate large-scale evaluation.

**Baselines.** We compare ARGOS with NuiScene (Lee et al., 2025). We evaluate two NuiScene variants: (1) the original model with its native $2 \times 2$ receptive field, and (2) a version we retrained with a maximum receptive field of $16 \times 16$ with mixed $2 \times 2$, $4 \times 4$, $8 \times 8$, and $16 \times 16$ multi-scale training. The latter ensures a fair comparison by matching the contextual window of our method. Since NuiScene does not support text conditioning, we evaluate our conditional generation results qualitatively. In addition, to decouple the contribution of our hierarchical AR framework from VAE improvements, we conduct a controlled experiment. Specifically, we train the ARGOS AR generator on the original frozen NuiScene VAE at its native resolution $(50, h_{vox}, 50)$, keeping all other settings unchanged.

**Evaluation protocol.** For unconditional generation, we employ Fréchet PointNet++ Distance (FPD) and Kernel PointNet++ Distance (KPD) (Qi et al., 2017) with 61440 sampled points per chunk. We evaluate at $2 \times 2$, $8 \times 8$, and $32 \times 32$ scales, generating 10k, 640, and 40 scenes respectively. For conditional generation, we report two quantitative metrics: (1) the CLIP score between input prompts and spatial annotations of the generated meshes, and (2) the cosine similarity between the text feature of the input prompt and the 3D feature of the point cloud sampled from the generated mesh extracted by the Uni3D model (Zhou et al., 2023). For this task, we use 100 diverse prompts from the test set, generating 8 scenes per prompt for scales of $2 \times 2$, $4 \times 4$, $8 \times 8$, and $16 \times 16$.

Table 3: Performance comparison between NuiScene variants (different receptive fields) and AR-GOS on unconditional generation. Note that KPD$^*$=KPD$\times 10^3$.

| Dataset | Method | 2×2 | | 8×8 | | 32×32 | |
|---|---|---|---|---|---|---|---|
| | | KPD$^*$ (↓) | FPD (↓) | KPD$^*$ (↓) | FPD (↓) | KPD$^*$ (↓) | FPD (↓) |
| 1 scene | NuiScene(2x2) (Lee et al., 2025) | **0.21** | 0.07 | 0.57 | 0.32 | 1.92 | 0.87 |
| | NuiScene(16x16) (Lee et al., 2025) | 0.22 | **0.06** | 0.54 | 0.29 | 1.38 | 0.76 |
| | ARGOS (**Ours**) | 0.24 | 0.07 | **0.52** | **0.24** | **0.91** | **0.71** |
| 13 scenes | NuiScene(2x2) (Lee et al., 2025) | 0.34 | **0.11** | 0.67 | 0.43 | 2.65 | 0.97 |
| | NuiScene(16x16) (Lee et al., 2025) | 0.34 | 0.12 | 0.63 | 0.34 | 2.36 | 0.82 |
| | ARGOS (**Ours**) | **0.33** | 0.12 | **0.59** | **0.27** | **1.97** | **0.75** |

Table 4: Architectural decomposition using frozen NuiScene VAE (single-scene). ARGOS AR framework outperforms NuiScene at large scales even with identical VAE.

| Method | 2×2 Chunks | | 8×8 Chunks | | 32×32 Chunks | |
|---|---|---|---|---|---|---|
| | FPD (↓) | KPD$^*$ (↓) | FPD (↓) | KPD$^*$ (↓) | FPD (↓) | KPD$^*$ (↓) |
| NuiScene (Original) | **0.049** | **0.11** | 0.37 | 0.48 | 1.26 | 1.33 |
| **ARGOS + NuiScene VAE** | 0.06 | 0.22 | **0.33** | **0.41** | **0.93** | **0.98** |

#### 4.3.2 RESULTS

**Unconditional Generation.** As shown in Tab. 3, ARGOS demonstrates competitive performance against NuiScene baselines. The advantage is most pronounced at larger scales ($8 \times 8$ and $32 \times 32$), demonstrating our method's superior ability to maintain global consistency. Qualitatively, Fig. 5

shows that ARGOS generates scenes with more organized layouts and finer geometric details. Notably, retraining NuiScene with a larger receptive field substantially improves its performance, validating the importance of a large contextual window for scene coherence. This strong performance holds on the more diverse 13-scene dataset, where metrics show a slight decrease as expected, reflecting the increased complexity. Additional visual comparisons are provided in A.6.

Table 4 presents the results on single-scene data. Despite identical reconstruction baselines, ARGOS outperforms the NuiScene baseline significantly at large scales ($32 \times 32$). While performance is comparable at small scales ($2 \times 2$), ARGOS exhibits superior scalability as scene complexity increases. Qualitative comparisons in Fig. 6 (13-scene data) further demonstrate that ARGOS generates coherent layouts with reduced repetitive patterns. These results confirm that our hierarchical design is the primary contributor to long-range coherence, independent of VAE fidelity.

**Conditional Generation.** Our method demonstrates robust and scalable spatial control via text conditioning. Tab. 5 shows high CLIP scores (0.98 at $2 \times 2$ to 0.91 at $16 \times 16$) and strong Uni3D similarities. The stable performance across scales underscores our model's ability to reliably maintain spatial constraints. As shown in Fig. 4, ARGOS excels at rendering complex relative arrangements, correctly preserving directional and height relationships even as the scene scales. These results validate our framework's effectiveness in multi-modal spatial control. Additional visual comparisons are provided in A.6.

Table 5: Quantitative evaluation for spatial-controlled large-scene generation. The CLIP and Uni3D scores serve to quantify text-geometry consistency and geometric plausibility, respectively. Their high values collectively demonstrate the statistical effectiveness and generalizability of the spatial control across multiple scales.

| Metrics | 2×2 | 4×4 | 8×8 | 16×16 |
|---|---|---|---|---|
| Clip score | 0.98 | 0.98 | 0.94 | 0.91 |
| Uni3D | 0.36 | 0.32 | 0.31 | 0.29 |

### 4.4 ABLATION STUDY

#### 4.4.1 VAE KEY MODULE ABLATIONS

ARGOS VAE mainly introduces two main techniques, which consist of flexible resolution control (FRC) and input with salient geometry guidance (SGG). Through our data collection pipeline, we can increase the occupancy supervision resolution from $(50, h_{vox}, 50)$ to $(256, h_{vox}, 256)$. We ablate the effectiveness of RFC and SGG in Tab. 6. We observe that high resolution oc-

Table 6: Ablations of RFC and SGG in ARGOS VAE. High supervision resolution and salient geometry guidance can both improve reconstructed results.

| Settings | | IoU (↑) | CD-$p$ (↓) | CD-$n$ (↓) | F-Score (↑) |
|---|---|---|---|---|---|
| FRC | SGG | | | | |
| × | × | 0.4690 | 0.0062 | 0.0212 | 0.8314 |
| ✓ | × | 0.6631 | 0.0009 | 0.0090 | 0.9657 |
| × | ✓ | 0.4692 | 0.0059 | 0.0225 | 0.8362 |
| ✓ | ✓ | **0.6686** | **0.0009** | **0.0087** | **0.9696** |

cupancy supervision and salient feature can both improve the reconstruct results but high-resolution supervision leads to a more substantial improvement. We also ablate the effectiveness of latent number in A.3 and find the number of latent (tested with 8, 16, 32, and 64) exhibits minimal impact on the reconstruction results, which can be attributed to the relatively simple geometry of the scene data. Consequently, a default latent dimension of 16 was adopted in this paper.

#### 4.4.2 UNCONDITIONAL GENERATION

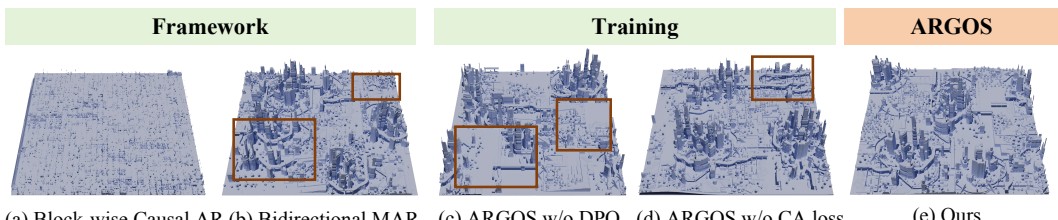

| Framework | | Training | | ARGOS |
|---|---|---|---|---|

(a) Block-wise Causal AR (b) Bidirectional MAR   (c) ARGOS w/o DPO   (d) ARGOS w/o CA loss       (e) Ours

Figure 7: Visual ablation of our hierarchical autoregressive method at 32x32 scale.

To efficiently validate our design, we conduct ablation studies on a single scene, with each chunk at a $(50, h_{vox}, 50)$ resolution. More visual results on 13 scene can be found in A.6.

Table 7: Ablation study of our hierarchical autoregressive method for unconditional generation.

| Method | 2×2 Chunks | | 8×8 Chunks | | 32×32 Chunks | |
|---|---|---|---|---|---|---|
| | FPD ($\downarrow$) | KPD ($\downarrow$) | FPD ($\downarrow$) | KPD ($\downarrow$) | FPD ($\downarrow$) | KPD ($\downarrow$) |
| Block-wise Causal AR | 0.13743 | 12.34 | 0.17264 | 14.57 | 0.21043 | 17.42 |
| Pure Bidirectional MAR | **0.00022** | 0.05 | 0.00038 | 0.13 | 0.00265 | 0.68 |
| w/o DPO | 0.00031 | 0.07 | 0.00072 | 0.16 | 0.00132 | 0.47 |
| w/o CA Loss | 0.00026 | 0.05 | 0.00043 | 0.13 | 0.00098 | 0.38 |
| **ARGOS (Full)** | 0.00025 | **0.05** | **0.00039** | **0.13** | **0.00095** | **0.36** |

**Hybrid Framework.** We compare our hierarchical design against two monolithic alternatives:

**(1) Block-wise Causal AR** model, which generates an entire chunk embedding in one step, performs the worst across all scales. As shown in Fig. 7 (a), its outputs are consistently coarse, suggesting that simultaneously modeling high-level structure and fine-grained detail is too complex for a single generative head.

**(2) Bidirectional MAR** model, which treats generation as a token-level masking task, is competitive on small 2×2 scenes. However, its performance degrades substantially on larger scales, producing repetitive and incoherent structures, as shown in Fig. 7 (b). Specifically, it highlights discontinuous element repetition (red box, bottom-left) and structural collapse in large scenes (top-right). This indicates that while effective for local synthesis, MAR struggles to maintain global consistency without explicit high-level guidance.

These results demonstrate the necessity of our hierarchical design, which successfully decouples global planning from local synthesis.

**Direct Preference Optimization (DPO).** A common failure mode in chunk-wise generation is the model defaulting to simplistic patterns like repetitive flat terrain or empty regions, as shown in Fig. 7 (c). We introduce DPO to mitigate this issue by encouraging structurally diverse outputs. As Tab. 7 shows, the inclusion of DPO significantly improves performance across all scales. It effectively steers the model away from low-complexity outputs and aligns its distribution with a preference for more structurally diverse scenes, proving essential for generating high-quality, large-scale results.

**Coherency-Aware Regularization.** Removing our Coherency-Aware Regularization (`w/o CA Loss`) consistently degrades performance across all scales (Tab. 7). Qualitatively, its absence introduces interruptions at chunk boundaries, accompanied by slight incoherent element repetition. Fig. 7 (d) shows incoherent repetition of "bridge" elements. This confirms its critical role in ensuring seamless transitions and preserving structural integrity.

## 5 CONCLUSION

We present ARGOS, a autoregressive framework that resolves the fundamental trade-off between global consistency and local fidelity in large-scale 3D scene generation. The key to our approach is a hierarchical decomposition: a causal autoregressive model first generates a global layout scaffold, which subsequently guides a masked autoregressive model in synthesizing high-fidelity geometric details. Our method not only sets a new state of the art in unconditional generation but also enables controllable synthesis, generating complex scenes that precisely adhere to spatial constraints specified in text. We believe ARGOS establishes a robust paradigm for virtual world generation, with promising future directions in accelerating inference and enabling richer semantic control.

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

## A  APPENDIX

### A.1  DATASET COLLECTION

**Scene mesh filter.** For fair comparison with NuiScene Lee et al. (2025), our dataset comprises 13 large-scale scenes featuring dense man-made structures, distinct from satellite-scanned data, as shown in Fig. 8. Ten scenes were manually selected from NuiScene43, supplemented with three urban scenes from public online repositories. To extend the generative context from 2×2 to larger configurations such as 8×8 and 16×16, a subset of the original mesh data was excluded due to the prevalence of extensive flat regions, which would otherwise limit the diversity and expressive capacity of the generative model.

**Scene mesh preprocessing.** To enhance data quality, raw meshes underwent a four-step manual processing pipeline: (1) removal of floating artifacts and extraneous regions; (2) normalization to the range $[-1, 1]$; (3) using architectural elements from a reference mesh as anchors to rescale meshes and ensure scale consistency; (4) ground plane regularization via occupancy grid conversion and flood-filling to detect the ground layer, followed by mesh bisection at the identified height and Boolean addition of a base cube to enforce a closed bottom geometry.

**Chunk data collection.** To extract chunks from preprocessed meshes, a sample mask is constructed to filter out regions with ground holes or excessive flatness as shown in Fig. 9. The scene mesh is converted to a signed distance field (SDF) and then to an occupancy grid, from which an alpha map and a depth variation map are derived for spatial filtering. The alpha map, formed by ground-plane projection, is convolved with a 100×100 unit kernel to identify low-occupancy regions as holes. The depth variation map is used to exclude areas with height variability below a set threshold. Qualified regions are segmented using Blender's bisect plane into 0.1×0.1 grid meshes. In single-scale experiments, a horizontal region of $[k - 0.05, k + 0.05]$ centered on each sample point is extracted; in multi-scale experiments, the region is partitioned into an $n \times m$ grid. Each slice grid is made watertight, from which SDF, occupancy, uniform and salient point clouds, and normals are extracted. Due to direct mesh splitting, occupancy is sampled at higher resolution than

702
703
704
705
706
707
708
709
710
711
712
713
714
715
716
717
718
719
720
721
722
723
724
725
726
727
728
729
730
731
732
733
734
735
736
737
738
739
740
741
742
743
744
745
746
747
748
749
750
751
752
753
754
755

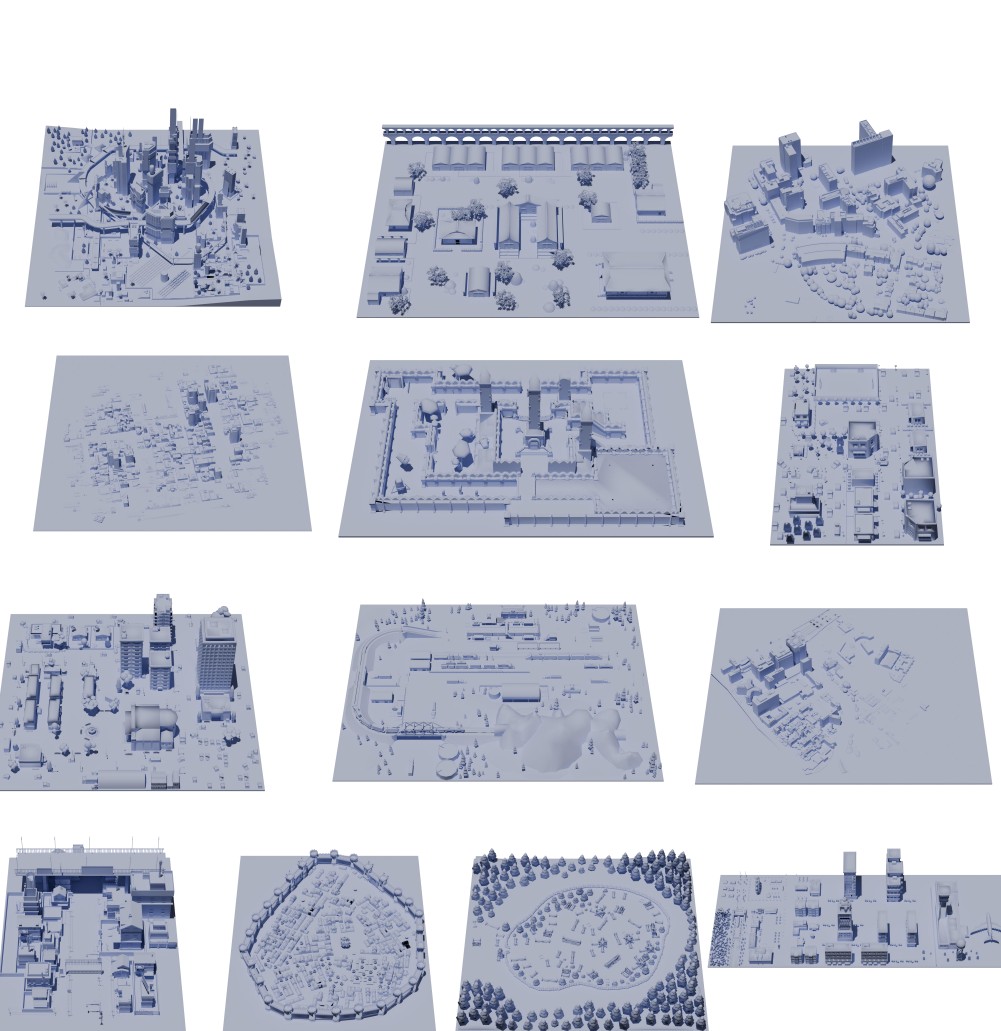

Figure 8: Raw mesh data for 13 scenes.

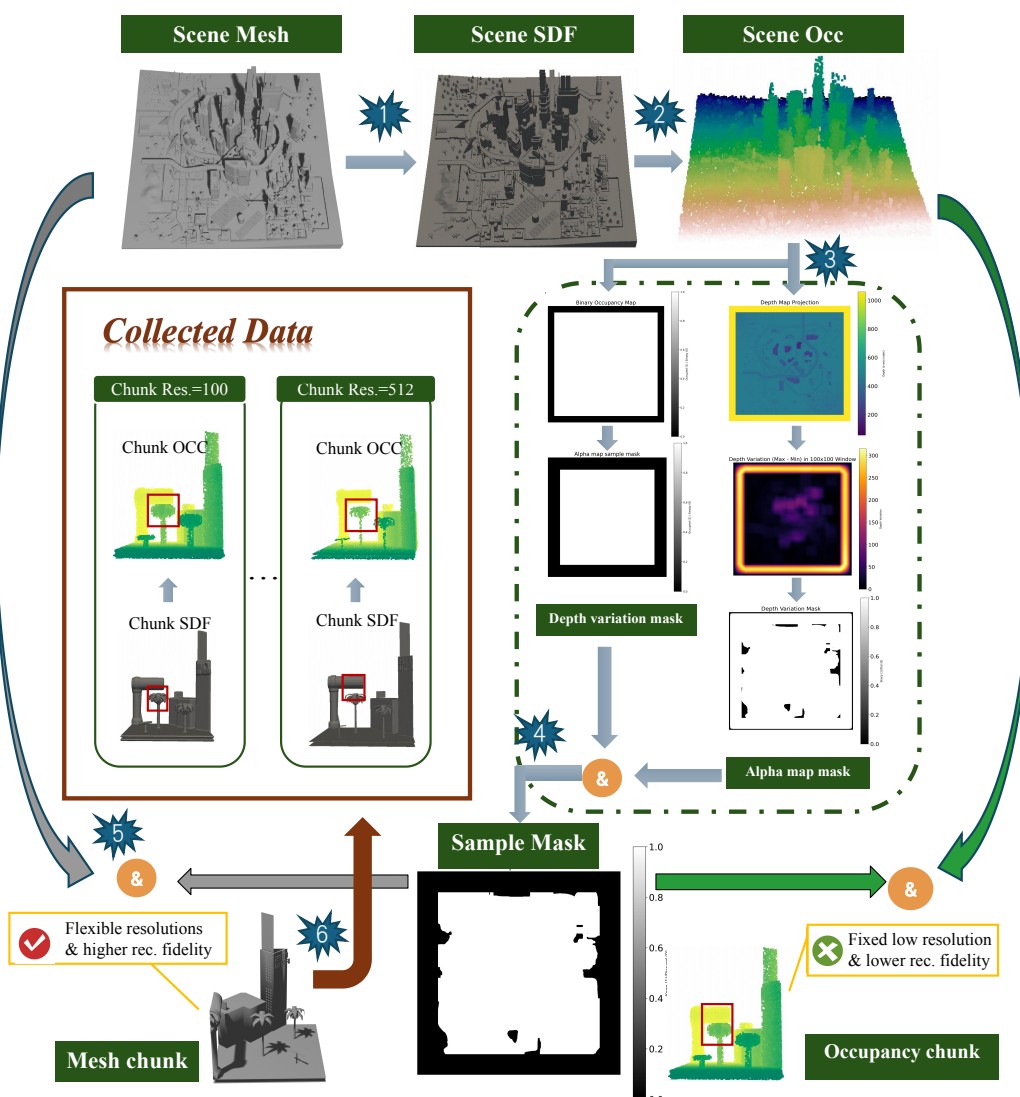

Figure 9: Illustration of chunk data collections.

NuiScene, yielding superior ground truth. Each single-scale slice is subdivided into 2×2 chunks, while multi-scale processing yields $n \times m$ such chunks, which are reordered for adjacent storage. A 90° counterclockwise rotation about the z-axis is applied to slice indices to align Blender and Trimesh coordinate conventions. Finally, we build onescene and 13 scene chunk dataset with various gird size of 2×2, 4×4, 8×8, 16×16 and 8×20 as shown in Tab. 1.

## A.2 PRELIMINARY: OCCUPANCY CHUNK VAE

Given a mesh representation of one scene, NuiScene converts the mesh to occupancy $\{0,1\}^{\mathcal{L} \times \mathcal{H} \times \mathcal{W}}$ and divides the whole occupancy into multiple small occupancy chunks with resolution $\{0,1\}^{50 \times h_{vox} \times 50}$ along the $l$ and $w$ axis, where $h_{vox}$ is the height of the corresponding chunk. Scene point cloud are uniformly sampled from the scene mesh origin from the scene occupacy. NuisScene further follows 3DShape2VecSet to build the chunk VAE. Specifically, the Chunk VAE incorporates three key modules for geometric reconstruction as follows.

**Geometry encoding.** Chunk VAE first slice the chunk point cloud $P$ from the scene point cloud. Then, Chunk VAE leverage cross-attention to aggregate $P_s$ and a learnable query set $L$, further obtains the latent vectors $Z_p$:

$$Z = \mathrm{CrossAttn}(L, \mathrm{PosEmb}(P))). \tag{11}$$

In addition, to mitigate the posterior collapse issue, Chunk VAE samples another point clouds $\hat{P}$ from the same chunk and constrains the similarity between $Z$ and $\hat{Z}$. Furthermore, based on $Z$, an additional branch of occupancy height prediction is introduced to improve the efficiency of occupancy prediction during inference.

**Geometry decoding.** Next, Chunk VAE decode $Z_p$ to occupancy with a sequential self-attention blocks and a vector set head as:

$$O = \mathrm{CrossAttn}(\mathrm{PosEmb(Q)}, \mathrm{SelfAttn}(Z)), \tag{12}$$

where $Q$ is the query points from the volume space $\mathcal{R}^{50 \times h_{vox} \times 50}$ and $O$ is the predicted occupancy. During inference, the final mesh is reconstructed with Marching Cubes.

## A.3 ABLATION OF LATENT NUMBER IN VAE ARCHITECTURE

We ablate the effectiveness of latent number and found the number of latent (tested with 8, 16, 32, and 64) exhibits minimal impact on the reconstruction results, which can be attributed to the relatively simple geometry of the scene data as shown in Tab.8. Consequently, a default latent dimension of 16 was adopted in this paper.

Table 8: Ablations of latent number.

| # Latents | IoU ($\uparrow$) | CD-$p$ ($\downarrow$) | CD-$n$ ($\downarrow$) | F-Score ($\uparrow$) |
|---|---|---|---|---|
| 8 | 0.6657 | 0.0009 | 0.0088 | 0.9669 |
| 16 | 0.6686 | 0.0009 | 0.0087 | 0.9696 |
| 32 | 0.6633 | 0.0009 | 0.0088 | 0.9651 |
| 64 | 0.6709 | 0.0009 | 0.0086 | 0.9721 |

## A.4 THE USE OF LARGE LANGUAGE MODELS (LLMS)

We employed a Large Language Model (LLM) as a writing assistant to enhance the clarity, precision, and readability of this manuscript. Our process involved providing author-written drafts to the LLM for suggestions on sentence restructuring, word choice, and grammatical corrections. All model-generated suggestions were critically reviewed, revised, and approved by the authors, who take full responsibility for the final content.

| Hyperparameter | Value |
|---|---|
| Optimizer | AdamW |
| Learning Rate | $1 \times 10^{-4}$ (constant) |
| Weight Decay | 0.02 |
| Adam $(\beta_1, \beta_2)$ | (0.9, 0.95) |
| Gradient Clipping | 3.0 |
| Total Epochs | 500 |
| Warmup Epochs | 100 |
| EMA Decay | 0.9999 |
| Condition Dropout | 0.1 |

Table 9: General training hyperparameters for ARGOS AR.

| Resolution | Batch Size (per GPU) |
|---|---|
| $2 \times 2$ | 64 ($4\times$ base) |
| $4 \times 4$ | 16 ($1\times$ base) |
| $8 \times 8$ | 4 ($0.25\times$ base) |
| $16 \times 16$ | 1 ($0.0625\times$ base) |
| $8 \times 20$ | 1 ($0.0625\times$ base) |

Table 10: Batch sizes for our multi-resolution training strategy.

### A.5 IMPLEMENTATION DETAILS

#### A.5.1 ARGOS VAE IMPLEMENTATION DETAILS

We describe the implementation details of ARGOS VAE from the architecture and training details. The main VAE architecture is illustrated as Fig. 1. The decoder has 24 self-attention layers. For VAE training with occupancy size $(50, h_{vox}, 50)$, the number of uniformly sampled point clouds, sharp-edge sampled point clouds, and query points is 4096. When increasing chunk resolution to $(256, h_{vox}, 256)$, we use 20480 input points and query points for VAE training. The loss ratio of $\lambda_{bce}$, $\lambda_{kl}$, $\lambda_{emb}$ and $\lambda_{height}$ is 1.0, 0.001, 1.0 and 1.0 respectively. All VAE results in this paper are obtained by training 160 epochs. We use LinearWarmupCosineAnnealingLR as learning scheduler and AdamW as the optimizer. The learning rate is $1e-5$ and the training batch size is 40. We trained ARGOS VAE with 8 A800 GPUs. It takes almost 1.5 days and 4 days to train VAE in the 1 scene and 13 scene dataset respectively.

#### A.5.2 ARGOS AR IMPLEMENT DETAILS

**Architecture.** ARGOS AR consists of three main components, as shown in Figure 2:

- Global Structure Generation. A causal transformer responsible for the global scene layout. It comprises 16 layers, 16 attention heads, and uses Rotary Position Embedding (RoPE). Causal masking is applied to ensure sequential generation. While our formulation (Eq. 4) models dependency on all previous tokens ($z_{<i}^{\text{global}}$), the effective context is determined by training data. Our largest training scale with sufficient samples is $16 \times 16$ chunks, and since our VAE compresses each chunk to $l = 16$ tokens, the model uses a maximum context of 4096 tokens ($16 \times 16 \times 16$). For scenes exceeding this size (e.g., $32 \times 32$), the model attends to the most recent 4096 tokens via a sliding window mechanism.

- Local Detail Generation. A masked autoregressive transformer for synthesizing fine-grained local details. It also has 16 layers and 16 attention heads but employs a random masking strategy for bidirectional context modeling within each chunk.

- Diffusion Head. A 12-layer MLP with a hidden dimension of 1536.

**Training Details** All models are trained using the AdamW optimizer on 8 NVIDIA H800 GPUs with Automatic Mixed Precision (AMP). Key hyperparameters are summarized in Table 9. Moreover, we employ a dynamic batching strategy based on generation resolution. The base batch size was 16 per GPU. Details are in Table 10. The total training time was approximately one week for single-scene datasets and two weeks for our full 13-scene dataset.

**Inference Details** Taking the $8 \times 8$ grid (64 chunks) as a representative benchmark, the total inference latency is 6 minutes (evenly decomposed into AR planning and VAE decoding) with a peak memory footprint of 17 GB.

## A.6 MORE QUALITATIVE RESULTS

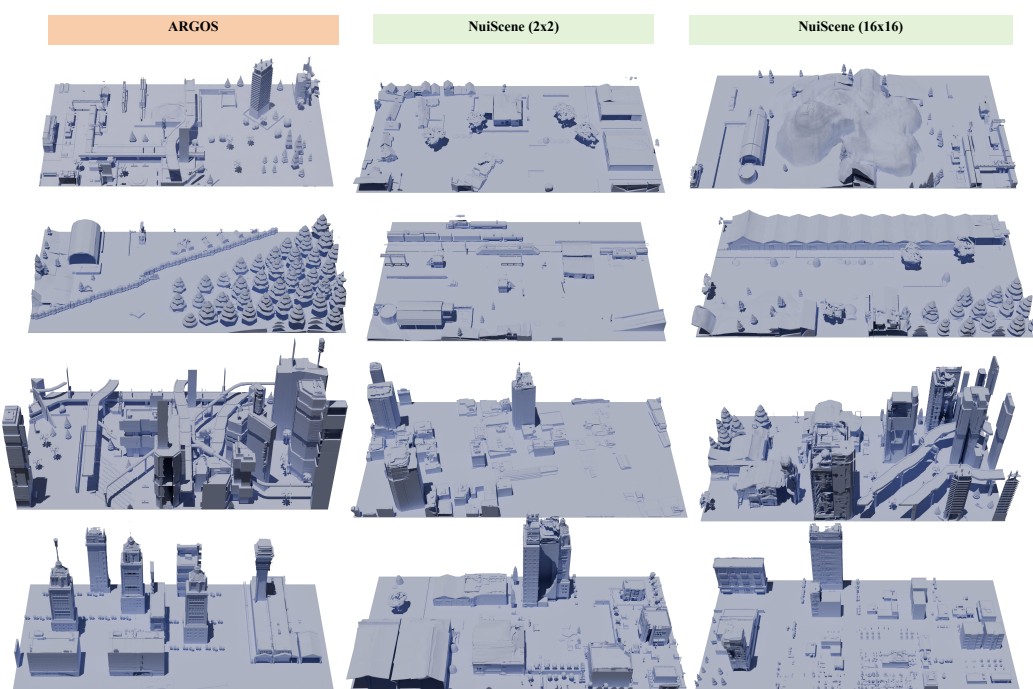

Figure 10: **Qualitative comparison of unconditional generation.** We compare ARGOS against the original NuiScene and the enlarged-receptive-field baseline. ARGOS demonstrates superior performance in long-range coherence, global planning, and geometric detail preservation.

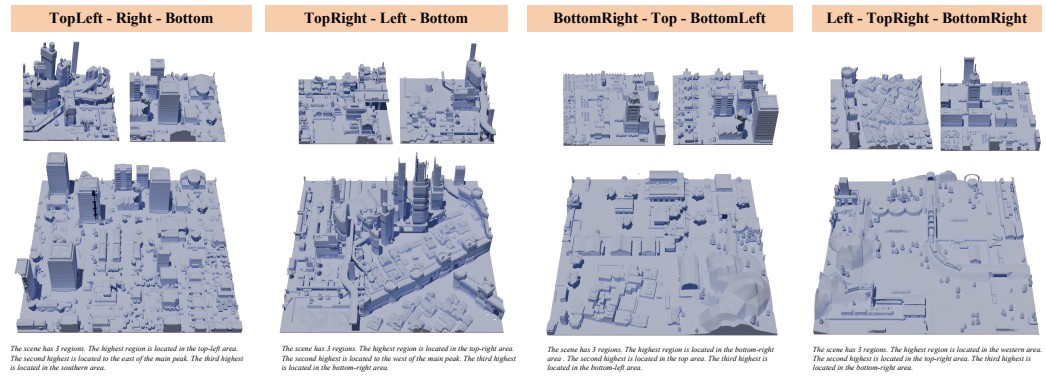

Figure 11: Conditional generation results across different spatial configurations at 8×8 and 16×16 resolutions. Top labels show simplified region orderings, bottom text provides full conditional descriptions.

**Unconditional Generation** Figure 10 presents qualitative comparisons against the original NuiScene and our re-implemented baseline with an enlarged receptive field. ARGOS demonstrates significant improvements across three key dimensions. First, unlike NuiScene which exhibits visible chunk boundaries and layout inconsistencies, our hierarchical framework maintains seamless spatial continuity across large scales. Furthermore, regarding, ARGOS generates holistic structures with logical spatial arrangements, whereas NuiScene often produces fragmented and abruptly transitioning layouts. Finally, our enhanced VAE reconstructs high-frequency details with sharper edges and fewer artifacts, effectively preserving architectural features.

**Conditional Generation**  Figure 11 illustrates results for four distinct prompts at multiple scales ($8 \times 8$ and $16 \times 16$), with three samples generated per prompt. The results validate the controllability and diversity of ARGOS. First, the model precisely follows spatial constraints. Beyond strict adherence, it exhibits geometric diversity, where significant variation is observed within constraints, ranging from diverse architectural styles to distinct terrain patterns. Finally, consistent instruction following is maintained from small to large scales. These results establish the superior quality and controllability of ARGOS.

**Visual Ablation on Unconditional Generation.**  To evaluate stability across random seeds, we visualize additional samples at $8 \times 8$ resolution on 13 scenes in Fig. 13. Bidirectional MAR (Fig. 13 (a)) fails to maintain long-range consistency, producing disordered arrangements and large-scale corruption. w/o DPO (Fig. 13 (b)) defaults to simplistic patterns with excessive empty space. w/o CA Loss (Fig. 13 (c)) exhibits boundary artifacts; the absence of coherency constraints causes geometric discontinuities, including repetitive elements and abrupt structural interruptions at chunk interfaces.

## A.7  MORE BASELINES

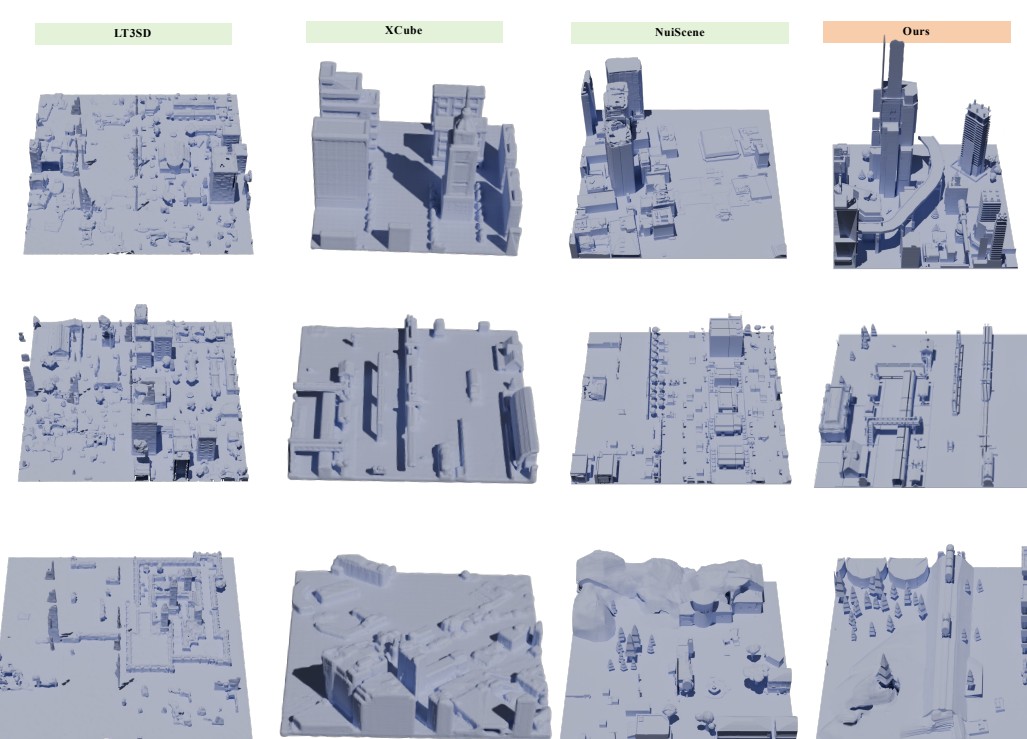

Figure 12: Qualitative comparison with baselines at $8 \times 8$ scale.

We have conducted a preliminary comparison with XCube (Ren et al., 2024b) and LT3SD (Meng et al., 2025).

### A.7.1  COMPARED WITH XCUBE

**Experimental Setting.**  We conduct the evaluation at a fixed $8 \times 8$ chunk scale, strictly following the requirement of XCube for resolution consistency between training and inference phases. Initial edge-prioritized sampling caused geometric holes. We retrained XCube using dense uniform sampling to ensure sufficient input information.

1026
1027
1028
1029
1030
1031
1032
1033
1034
1035
1036
1037
1038
1039
1040
1041
1042
1043
1044
1045
1046
1047
1048
1049
1050
1051
1052
1053
1054
1055
1056
1057
1058
1059
1060
1061
1062
1063
1064
1065
1066
1067
1068
1069
1070
1071
1072
1073
1074
1075
1076
1077
1078
1079

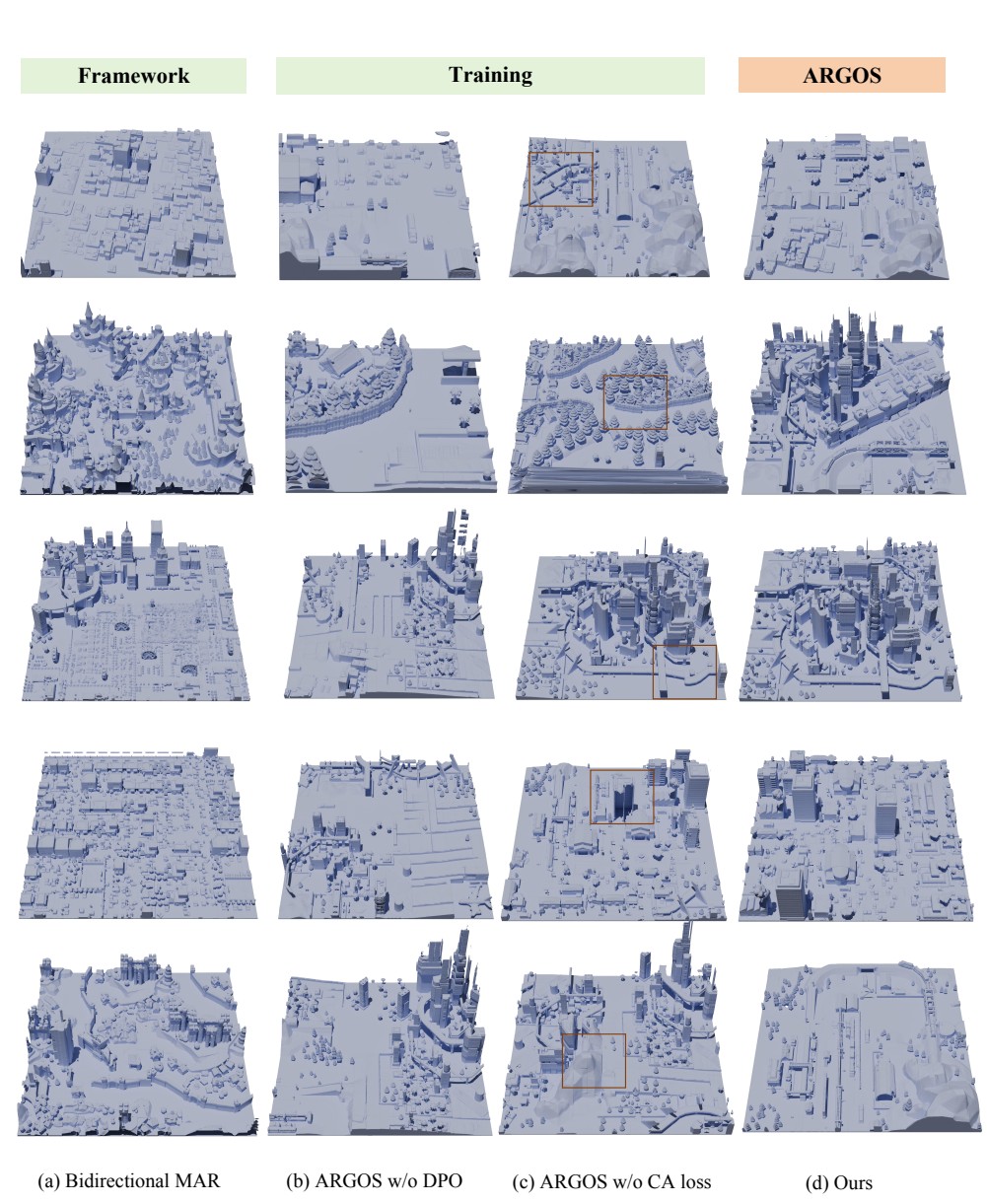

| Framework | Training | ARGOS |
|---|---|---|

(a) Bidirectional MAR    (b) ARGOS w/o DPO    (c) ARGOS w/o CA loss    (d) Ours

Figure 13: Qualitative ablation on $8 \times 8$ unconditional generation. We present five random samples per variant to visualize stability. (a) Bidirectional MAR lacks global guidance, resulting in cluttered layouts and occasional structural collapse (Row 3). (b) w/o DPO suffers from mode collapse, exhibiting a high proportion of empty regions. (c) w/o CA Loss introduces geometric discontinuities at chunk boundaries, manifesting as element repetition (Row 1) or unnatural structural interruptions.

**Results.** As shown in Fig. 12, the retrained XCube produces reasonable structures. However, its global embedding VAE formulation struggles to preserve fine details over the large spatial extent of an $8 \times 8$ scene compared to our chunk-based approach. Additionally, the strict alignment between training and inference resolutions limits its flexibility for large-scale extrapolation.

### A.7.2 COMPARED WITH LT3SD

**Experimental Setting.** LT3SD is primarily evaluated on the indoor dataset 3D-FRONT. Following the original implementation, we preprocess our 13 outdoor mesh samples and train the LT3SD model accordingly. The voxel size is set to 1.8 meters, with the maximum height of each mesh constrained to under 256 resolution. Apart from the dataset adaptation, we maintain nearly identical training and inference configurations as the original codebase. The LT3SD VAE is trained for 100K iterations, while the diffusion model undergoes 800K iterations of training.

**Results.** We extend our gratitude to the LT3SD authors for their valuable suggestions that facilitated the reproduction of normal results as shown in Fig.12. Through visual inspection of the results, we observe that the scene geometry shows noticeable degradation compared to our ARGOS. Architectural structures and vegetation appear blurred with significant floating artifacts, while the spatial layout demonstrates limited coherence. We attribute the geometry quality decline primarily to resolution constraints. Similar to NuiScene, LT3SD employs voxelization of entire scenes followed by online chunking, which inherently restricts chunk shapes due to voxel conversion parameters. Concerning the spatial incoherence, we hypothesize this stems from the same fundamental limitation as in NuiScene: the outpainting process lacks comprehensive global guidance, resulting in locally optimal but globally suboptimal arrangements.

### A.8 TEXTURE BAKER RESULTS

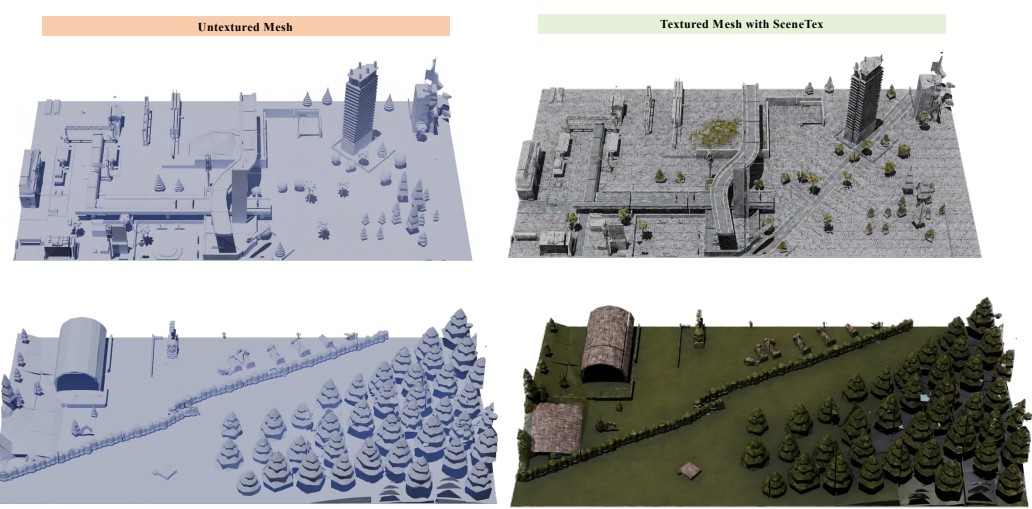

Figure 14: Texture baker results with SceneTex.

We add two texture baker visualization results, as shown in Fig. 14. SceneTex (Chen et al., 2024) is adopted as the texture generation model. For each untextured mesh, we normalize each scene to $[-1, 1]$ and sample 1.6K camera poses in blender platform to build the camera trajectory. Each scene is optimized with H100 for about 12 hours.

### A.9 LIMITATIONS

For conditional generation, our approach leverages text to guide spatial arrangements, yet natural language is inherently under-constrained for defining complex 3D geometry. Although our automated pipeline mitigates this ambiguity via structured topological constraints, a fundamental trade-

off persists between prompt flexibility and geometric determinism. Consequently, text alone proves insufficient for scenarios demanding strict metric adherence. We envision addressing this limitation by integrating hybrid control signals, fusing the semantic richness of text with the explicit precision of bounding boxes or segmentation maps.

## A.10 REPRODUCIBILITY STATEMENT

We commit to release our code, model and dataset for reproducibility.

