# OpenReview forum: "ARGOS: Hierarchical Autoregressive Generation of Unbounded 3D Outdoor Scenes with High Fidelity and Spatial Control"
_ICLR.cc/2026/Conference — Submitted to ICLR 2026_

### Official Review · Reviewer_aDSy · 2025-10-22

**Soundness:** 2
**Presentation:** 1
**Contribution:** 2
**Rating:** 2
**Confidence:** 4

**Summary:**

This paper proposes a hierarchical autoregressive model for 3D outdoor synthesis. Specifically, a causal autoregressive model is designed for globally coherent layout generation and a masked autoregressive model for conditional detailed local geometry generation. Experimental results have shown improved performance on FPD and KPD compared to existing methods.

**Strengths:**

+ The qualitative results and quantitative results shown are better than existing models for outdoor scene generation.
+ Summary of related works are sufficient.

**Weaknesses:**

The novelty of the paper overall is not significant. The paper seems to be a direct replacement of diffusion model in NuiScene with autoregressive model. In details:
1. The proposed VAE compared to the VAE in NuiScene is simply an additional dense surface sampling for high fidelity geometry.
2. Unclear motivation for the design of $\textbf{L}_{height}$.
3. The motivation of using an autoregressive model rather than diffusion model is unclear.
4. The motivation of the hierarchical AR model rather than a direct next token prediction of 3D grid.
5. Missing ablation study on visual results.
6. Unclear explanation of 13 scene data for training. What advantage does 13 scenes brings?
7. Missing ablation study on $\lambda_{coh}$ and $\lambda_{dpo}$.
8. There is only one example (Figure 5) that shows the performance compared to existing method. More results should be provided to validate the design of the model. Additionally, the visual results does not show improved results than NuiScene.
9. Diversity was not shown.

**Questions:**

See Weakness

---

> ### Author Response · Authors · 2025-11-23
>
> We thank the reviewers for their constructive feedback. We address the key questions and concerns raised by the reviewers as follows.
>
> **W1: The proposed VAE compared to the VAE in NuiScene is simply an additional dense surface sampling for high fidelity geometry.**
>
> **A1:** We acknowledge that from an architectural perspective, both ARGOS VAE and NuiScene VAE largely follow the framework of 3DShape2VectorSet, with minimal structural differences. Our key improvements over NuiScene VAE lie in the model's input and supervision signals. Our experiments demonstrate that incorporating sharp-edge sampled point clouds as input and increasing the occupancy resolution significantly enhances reconstruction quality. This is particularly true for the increase in occupancy resolution, as evidenced by the results in Table 6 of revision. However, scaling to higher chunk resolutions is inherently constrained by the mesh2sdf framework underlying NuiScene's original occupancy collection pipeline as dicussed in Sec 3.1.
>
> **W2: Unclear motivation for the design of $L_{height}$**
>
> **A2:** Actually, we follow the design of $L_{height}$ from NuiScene. These outdoor scenes have varying heights and the height prediction is used to prune unnecessary occupancy predictions during inference time.
>
> **W3: Motivation of Using Autoregressive Model Rather Than Diffusion Model**
>
> **A3:** We adopt Autoregression (AR) over Diffusion to ensure global consistency in unbounded generation. Diffusion methods, typically relying on local inpainting windows, lack an explicit memory mechanism to propagate spatial history beyond their immediate context. This deficiency leads to semantic drift and boundary artifacts as the scene expands. In contrast, Causal AR explicitly conditions on the ​compressed global history, enforcing logical consistency across the full sequence. However, applying vanilla AR directly to raw tokens compromises local fidelity (as shown in Table 7, where "Block-wise Causal AR" yields coarse outputs). Our Hierarchical Design resolves this by decoupling global planning (Causal AR) from local synthesis (Masked AR), thereby reconciling long-range consistency with high-frequency geometric detail.
>
> **W4: Motivation of Hierarchical AR Rather Than Direct Next-Token Prediction of 3D Grid.**
>
> **A4:** Hierarchical decomposition is necessary because direct token prediction faces an ​optimization bottleneck​. Simultaneously resolving large-scale structure and high-frequency detail in a single stream is optimizationally intractable. Our ablations (Table 7 in revision) confirm this failure mode: single-stage "Block-wise Causal AR" yields overly coarse geometry, whereas "Pure Bidirectional MAR" collapses into repetitive patterns at scale. Our architecture aligns the modeling paradigm with the data structure: the Global Stage (Causal Z-order) establishes spatial coherence through sequential dependency, while the Local Stage (Masked AR) models the unordered geometric tokens conditioned on this layout.  This separation enables the optimization of both global coherence and local fidelity, a result unattainable by single-stage baselines.
>
> **W5: Missing ablation study on visual results.**
>
> **A5:** We clarify that qualitative ablations are explicitly presented in the manuscript.
>
> - **For AR generation**, Figure 7 (in revision) provides visual comparisons at the $32 \times 32$ scale, contrasting our hierarchical approach against single-stage baselines (Block-wise Causal and Pure MAR). Annotated regions highlight key differences in generation quality, such as boundary coherence and structural organization. These visual results are supported by quantitative ablations in Table 7 (in revision).
>
> - **For VAE reconstruction**, visual comparisons are provided in Figure 3 of main paper, demonstrating that ARGOS VAE produces finer geometric details compared to NuiScene VAE across both single-scene and 13-scene datasets.
>
> **W6: Unclear explanation of 13 scene data for training. What advantage does 13 scenes brings?**
>
> **A6:** To ensure a fair comparison with NuiScene, we consequently selected and trained our model on 13 scenes sourced from both NuiScene43 and Objaverse. During this curation process, we found that many scenes suffered from poor quality, such as containing large blank or empty regions. We fully acknowledge that the scarcity of high-quality 3D data significantly limits the generalization capability of generative models, which is a key direction for our future work.

---

> > ### Author Response · Authors · 2025-11-23
> >
> > **W7: Missing ablation study on $\lambda_{coh}$ and $\lambda_{dpo}$**
> >
> > **A7:** These coefficients were selected to balance all loss terms at similar magnitudes during training. We conducted sensitivity analyses on a subset of single-scene data at the 32x32 scale to validate the hyperparameters $\lambda_{coh} = 0.1$ and $\lambda_{dpo} = 0.05$.
> >
> > Table 1: Ablation Study on Coherency Loss Weight (λ_coh). We fix λ_dpo = 0.05 and vary λ_coh at 32×32 scale on single-scene data. KPD values are scaled by 10³.
> >
> > | λ_coh | FPD (↓) | KPD* (↓) |
> > |-------|---------|----------|
> > | 0.05  | 0.00072    | 0.44     |
> > | **0.10** | **0.00071** | **0.41** |
> > | 0.20  | 0.00074    | 0.48     |
> >
> > Table 2: Ablation Study on DPO Loss Weight (λ_dpo), We fix λ_coh = 0.1 and vary λ_dpo at 32×32 scale on single-scene data. KPD values are scaled by 10³.
> >
> > | λ_dpo | FPD (↓) | KPD* (↓) |
> > |-------|---------|----------|
> > | 0.025 | 0.00073    | 0.46     |
> > | **0.05** | **0.00071** | **0.41** |
> > | 0.10  | 0.00072    | 0.43     |
> >
> > The results indicate that performance is stable around our chosen parameters, confirming the robustness of the framework to minor hyperparameter variations.
> >
> > **W8: More results should be provided to validate the design of the model and diversity are not shown. Additionally, the visual results does not show improved results than NuiScene.**
> >
> > **A8:**  We have enriched the manuscript with expanded qualitative comparisons, refer to Sec A.6:
> >
> > - **Unconditional Generation:**  We provide new multi-scene visualizations contrasting ARGOS against NuiScene. These results explicitly illustrate the long-range coherence of ARGOS, effectively resolving the fragmented transitions and boundary artifacts observed in the baseline.
> > - **Conditional Generation:**  We present generation results for 4 diverse prompts (3scenes per prompt). These samples verify that ARGOS achieves geometric diversity, while maintaining rigorous adherence to the prescribed spatial constraints.

---

> > ### Comment · Reviewer_aDSy · 2025-11-26
> > **Comment on A5**
> >
> > Hi, thank you for your response. I am confused about the ablation study shown in Figure 7. Why do you consider the marked regions unsatisfying? I do not sense the boundary inconsistency there. Besides, could you show a few more examples, say 5-10 for visual comparion?

---

> ### Author Response · Authors · 2025-11-28
> **Response to Visual Ablation**
>
> We thank the reviewer for the feedback on the visual ablation results.
>
> - **Visual Ablation Analysis**
>   Since the failure of Block-wise Causal AR is evident, we focus on **Bidirectional MAR**, **w/o DPO**, and **w/o CA (Coherency-Aware) loss**. Beyond Fig.7 in the main paper, we provide additional $8\times8$ results (5 samples per model) in Fig. 13 of the revised manuscript to visualize stability.
>   - **Bidirectional MAR:** Lacks explicit global guidance, preventing long-range consistency. Fig.7(b) highlights discontinuous element repetition (red box, bottom-left) and structural collapse in large scenes (top-right). Fig.13(a) confirms this, showing cluttered layouts and large-scale corruption (Row 3).
>   - **w/o DPO Loss:** Suffers from mode collapse. The model favors simplistic patterns, such as repetitive flat ground or substantial empty space. This is marked in Fig.7(c) and evident in Fig.13(b), which exhibits a larger proportion of empty regions.
>   - **w/o CA Loss:** Addresses relatively subtle consistency issues, as reflected in the quantitative metrics of Table 7. Its absence introduces interruptions at chunk boundaries, accompanied by slight incoherent element repetition. Fig.7(d) shows incoherent repetition of "bridge" elements. Fig.13(c) illustrates similar artifacts: Row 1 highlights element repetition and subsequent rows show unreasonable structural interruptions.

---

### Official Review · Reviewer_hjzj · 2025-10-31

**Soundness:** 2
**Presentation:** 4
**Contribution:** 3
**Rating:** 6
**Confidence:** 4

**Summary:**

ARGOS is introduced as a hierarchical autoregressive generation framework that excels at large-scale outdoor scene generation. It also supports text-based conditional generation. Based on previous framework from NuiScene, it introduces a two-stage generation process: a global layout generation stage that leverages causal autoregressive modeling to create coherent layouts, followed by a second autoregressive modeling that fills in fine details at each local chunk. ARGOS also leverages a new VAE design for better local detail reconstruction compared to NuiScene. The authors claim ARGOS achieves superior performance in unconditional generation and shows promising results in text-conditional generation.

**Strengths:**

1. Introducing a hierarchical autoregressive framework for large-scale outdoor scene generation is intuitive and well-motivated. It brings solid improvements over large scene coherence compared to prior work with diffusion outpainting.
2. New VAE design with mesh voxelization and salient geometry guidance brings better resolution and detail reconstruction.
3. Solid empirical comparison with NuiScene, retraining NuiScene with larger receptive field adds credibility to the comparison.

**Weaknesses:**

- **Major:**
1. The 13-scene dataset processed by the authors is sufficiently large for training and in-context evaluation. However, compared to the full NuiScene43 dataset and the original Objaverse data, the lack of diversity in only 13 scenes may limit the generalizability to other outdoor scenes. I did not see any evaluation of ARGOS on data outside of the 13-scene dataset.
2. The evaluation is primarily constrained to authors's own data pipeline, this does not tell the full story if ARGOS can still work significantly better than NuiScene using arbitrary reconstruction that wasn’t meshed and chunked this way.
3. The discussion of Related Work is limited. The authors claim there's limited work for structurally controllable outdoor scene generation while many works have explored this area, such as [1, 2, 3, 4, 5]
4. Following that, text-based signal is fundamentally unsuitable for complex scene control. Linguistic prompts are inherently ambiguous and struggle to provide precise spatial signal needed for complex scene generation. Example prompts in the manuscript such as “highest region in the top-right…” works because the authors derives a data-specific structure from their data pipeline. This strategy may not work for all scenarios. The authors should discuss this limitation more thoroughly.

- **Minor:**
1. I would like to see more qualitative comparisons beyond the scene style showed in the current manuscript. It would also be nice if the authors include their full dataset visualizations in the appendix for better understanding of the data distribution.
2. Label text color in Fig 2's global and neighbors latent blocks are hard to read against the background. Please change to a darker color.
3. Add a short reproducibility statement about code, model and dataset release plan.
4: The authors claim ARGOS has superior performance in unconditional generation, while it only significantly out performs NuiScene in larger scales, please change the wording to "ARGOS has superior performance over previous methods at large scale scene generation".

[1] UrbanWorld: An Urban World Model for 3D City Generation

[2] Text2LiDAR: Text-guided LiDAR Point Cloud Generation via Equirectangular Transformer

[3] Controllable 3D Outdoor Scene Generation via Scene Graphs

[4] InfiniCube: Unbounded and Controllable Dynamic 3D Driving Scene Generation with World-Guided Video Models

[5] SSEditor: Controllable Mask-to-Scene Generation with Diffusion Model

**Questions:**

Refer to Weaknesses.

---

> ### Author Response · Authors · 2025-11-23
>
> We thank the reviewer for the positive assessment of our training scale and in-context evaluation. We address the concerns regarding generalization, framework decoupling, and comparisons below.
>
> **W1: The 13-scene dataset processed by the authors is sufficiently large for training and in-context evaluation. Compared to the full NuiScene43 dataset and the original Objaverse data, the lack of diversity in only 13 scenes may limit the generalizability to other outdoor scenes. I did not see any evaluation of ARGOS on data outside of the 13-scene dataset.**
>
> **A1:** We thank the reviewer for their positive recognition of our training data volume and in-context evaluation, as well as for their valuable consideration regarding the scale of 3D scene data. While NuiScene released a 43-scene dataset (NuiScene43), their paper only reports results from models trained on 4 scenes. Although they provided an open-source model trained on 13 scenes, they did not actually train a model on the full NuiScene43. To ensure a fair comparison with NuiScene, we consequently selected and trained our model on 13 scenes sourced from both NuiScene43 and Objaverse. During this curation process, we found that many scenes suffered from poor quality, such as containing large blank or empty regions. We fully acknowledge that the scarcity of high-quality 3D data significantly limits the generalization capability of generative models, which is a key direction for our future work. Regarding the evaluation on data outside these 13 scenes, it is inherently challenging for ARGOS. Since our method is designed for unconditional generation or text-conditioned generation based solely on spatial layout, it is hard to adjusted to the conventional "out-of-domain" test.
>
> **W2: If ARGOS can still work significantly better than NuiScene using arbitrary reconstruction that wasn’t meshed and chunked this way.**
>
> **A2:** We explicitly decouple our hierarchical framework from the specific preprocessing pipeline.
>
> - **Architecture as a General Prior.** The core mechanism, Global Causal AR for layout planning followed by Local Masked AR for detail synthesis, operates strictly on ​latent token sequences​. This design is agnostic to the underlying 3D representation. Whether the tokens represent Occupancy grids, NeRF latents, or Gaussian groups is immaterial to the autoregressive formulation. The framework requires only that the scene be spatially decomposable. The VAE functions solely as a modular tokenizer, not a structural dependency.
> - **Empirical Verification.** For stricter comparison, we conduct a controlled experiment. Specifically, we train the ARGOS AR generator on the original frozen NuiScene VAE at its native resolution\~(\$50, h\_{vox}, 50\$), keeping all other settings unchanged.
> **Table: Quantitative Comparison using Frozen NuiScene VAE (Single-Scene)**
>
> | Method | 2×2(↓) | 8×8(↓) | 32×32(↓) |
> |--------|-----|-----|-------|
> |        | FPD/KPD* | FPD/KPD* | FPD/KPD* |
> | NuiScene original| **0.049/0.11** | 0.37/0.48 | 1.26/1.33 |
> | **ARGOS + NuiScene VAE** | 0.06/0.22 | **0.33/0.41** | **0.93/0.98** |
>
> Above table presents the results on single-scene data. Despite identical reconstruction baselines, ARGOS outperforms the NuiScene baseline significantly at large scales (32 x 32). While performance is comparable at small scales (2 x2), ARGOS exhibits superior scalability as scene complexity increases. Qualitative comparisons( refer to Figure 6 in revision manuscript) (13-scene data) further demonstrate that ARGOS generates coherent layouts with reduced repetitive patterns. These results confirm that our hierarchical design is the primary contributor to long-range coherence, independent of VAE fidelity.
>
> **W3: The discussion of Related Work is limited.**
>
> **A3:** Thanks for your kind remind. We have added more related work UrbanWorld、Text2LiDAR、Controllable 3D Outdoor Scene Generation via Scene Graphs、InfiniCube、SSEditor and CommonScenes、EchoScene、InstrucLayout mentioned by **Reviewer 1T4i** and discussed their relationship comprehensively in the related work section of the revised manuscript.
>
> **W4: Limitations of Text-Based Control**
>
> **A4:** We clarify the scope of our text-driven mechanism. Natural language is fundamentally coarse-grained, making it ​ill-posed for precise metric specification​. Consequently, our approach prioritizes high-level topological planning over geometric exactness. This design choice is validated by our automated pipeline (Sec. 3.2.3), which effectively leverages structured constraints to achieve strong spatial alignment (CLIP scores 0.91--0.98 in Table 5). While sufficient for layout prototyping, applications demanding rigorous adherence necessitate hybrid signals (e.g., bounding boxes) to complement semantic text. We have expanded the discussion of these modality-specific limitations in the revised manuscript.

---

> > ### Author Response · Authors · 2025-11-23
> >
> > **W5: more qualitative comparisons and full dataset visualizations.**
> >
> > **A5:** We have enriched the manuscript with expanded qualitative comparisons, refer to Sec A.6:
> >
> > - **Unconditional Generation:**  We provide new multi-scene visualizations contrasting ARGOS against NuiScene. These results explicitly illustrate the long-range coherence of ARGOS, effectively resolving the fragmented transitions and boundary artifacts observed in the baseline.
> > - **Conditional Generation:**  We present generation results for 4 diverse prompts (3scenes per prompt). These samples verify that ARGOS achieves geometric diversity, while maintaining rigorous adherence to the prescribed spatial constraints.
> >
> > To further understand the data distribution, we add full dataset visualizations in the Appendix (refer to Figure 8).
> >
> > **W6: Label text color in Fig 2's global and neighbors latent blocks are hard to read against the background.**
> >
> > **A6:**
> > Thank you for this feedback. We have revised Figure 2 to improve text legibility.
> >
> > **W7: Add a short reproducibility statement about code, model and dataset release plan.**
> >
> > **A7:**  We commit to release our code, model and dataset and have added the reproducibility statement to the Appendix of the revised manuscript.

---

### Official Review · Reviewer_1T4i · 2025-10-31

**Soundness:** 3
**Presentation:** 3
**Contribution:** 2
**Rating:** 6
**Confidence:** 3

**Summary:**

The paper proposes a hierarchical autoregressive pipeline for large-scale scene synthesis. A global causal model generates chunk-level layout in Z-order, and a local masked model with a diffusion head fills fine geometry per chunk. For controllability, the authors develop an automated text description pipeline and utilize CLIP text embeddings for conditioning.

**Strengths:**

1. The two-stage design, containing global causal over chunks and local masked autoregressive generation, directly targets coherence across large grids. DPO loss is introduced in the paper, which is nice to see, and helps to stabilize large-scale synthesis and diversity.

2. The paper is clearly written and easy to follow.

**Weaknesses:**

1.  3D chunk VAE is not technically more novel or has more differences than NuiScene.

2. Most comparisons are against NuiScene. Given the claim of outperforming “existing methods,” the paper should also compare with other recent large-scene or chunked methods.

3. The dataset is a bit narrow, as it only contains 13 scenes. This has an overfit risk, especially for the text-to-scene problem.

4. The paper details training setups, but does not systematically report inference latency, memory, or throughput for large grids.

5. L134-135 names two techniques, FRC and SGG, and refers to Figure 1. Yet, they are not drawn to it. The current way of designing the pipeline figure is no different from NuiScene.

6. Figure 2 needs a more obvious Z-order direction to be clearer.

7. Some work on controllable scene generation is missing in L93-94, e.g., CommonScenes, EchoScene, etc.

**Questions:**

Please see the weakness above.

Why did the authors just choose the adjacent top chunk and the left chunk for generating local details? How to define this window size? For a given chunk, conditioning can extend beyond the top and left neighbors to include diagonal context, such as the top-left and top-right chunks, for instance.

Speaking to this, why did the authors choose to use the z-order rather than the masked strategy for global chunk generation (see MaskGIT, for example)?

It would be better if the authors could demonstrate some texture baking for their examples.

---

> ### Author Response · Authors · 2025-11-23
>
> We thank the reviewer for their detailed assessment. We address the questions regarding novelty, baselines, and architectural choices below.
>
> **W1: 3D chunk VAE is not technically more novel or has more differences than NuiScene.**
>
> **A1**: While the ARGOS VAE and NuiScene VAE share a similar core architecture based on 3DShape2VectorSet, our primary enhancements pertain to the input data and supervision. Experiments in Tables 2 and 5 confirm that employing sharp-edge sampled point clouds and higher-resolution occupancy supervision substantially boosts reconstruction quality. Crucially, achieving such high resolution is a key advantage of our method, as the conventional mesh2sdf approach used in NuiScene poses a critical barrier to scaling as discussed in Sec.3.1. Our pipeline detailed in Sec A.1 collects occupancy with higher resolution~(50 vs 256) from chunked mesh rather than scene occupancy and the overcomes this limitation, providing a more novel and effective solution for high-fidelity supervision.
>
> **W2: More comparisons with other recent large-scene or chunked methods.**
>
> **A2:** We appreciate the suggestion to benchmark against EarthCrafter, XCube, and LT3SD. As EarthCrafter is not open-source, we provide comparisons with XCube and LT3SD.
> - **XCube:**
>   - **Setup:** We conduct the evaluation at a fixed 8 x 8 chunk scale, strictly following the requirement of XCube for resolution consistency between training and inference phases.
>   - **Results:** We present results derived from the official XCube preprocessing, as shown in Figure 12 of revision manuscript. These outputs exhibit geometric holes on ground planes, empirically indicating that XCube is inherently optimized for bounded objects rather than continuous outdoor surfaces. Specifically, the sampling routine prioritizes geometric edges, resulting in insufficient density on large planar regions . Consequently, this sparsity directly induces the observed artifacts. To ensure a robust baseline, we are currently retraining XCube with adapted dense uniform sampling and will report the optimized results.
> - **LT3SD:**
>   - **Setup:** LT3SD is mainly conducted in the indoor dataset 3DFront and we follow its codebase to preprocess our ourdoor mesh data and train the LT3SD model. We consider mesh chunk with 8x8 chunk size as the scene shape and sample 5K mesh chunk for LT3SD training. Each mesh chunk is processed with resolution 512³ .
>   - **Results:** We have obtained a coarse result, but the result seems strange (refer to Figure 12 in revision manuscript). We are currently investigating the issue and are unsure whether it stems from data processing problems or training and inference configuration issues. The Lt3sd code was primarily designed for indoor scene generation, which typically involves limited height variations, whereas outdoor scenes require handling diverse elevation changes and would likely need an additional adaptation process. In any case, we are striving to achieve normal results with Lt3sd on our dataset to enable a comparative analysis.
>
>
> **W3: The dataset is a bit narrow, as it only contains 13 scenes. This has an overfit risk, especially for the text-to-scene problem.**
>
> **A3:** We address the dataset concern from two perspectives:
>
> - **Controlled Benchmarking:** We adhere to the 13-scene setup to establish a rigorous baseline against NuiScene. Aligning the training data is requisite to attribute performance gains strictly to our architectural innovations rather than data scaling factors.
> - **Chunk-Level Generalization:** Our model learns geometric priors at the ​chunk level​. Consequently, the effective training set comprises tens of thousands of discrete samples (Table 1 in main paper), rather than 13 global instances. This magnitude provides sufficient variation to prevent the memorization of scene-level templates, ensuring robustness in generation tasks.
>
>
> **W4: The paper details training setups, but does not systematically report inference latency, memory, or throughput for large grids.**
>
> **A4:** Taking the $8 \times 8$ grid (64 chunks) as a representative benchmark, the total inference latency is 6 minutes (evenly decomposed into AR planning and VAE decoding) with a peak memory footprint of 17 GB. We have added the inference statistics in revision.
>
> **W5: L134-135 names two techniques, FRC and SGG, and refers to Figure 1. Yet, they are not drawn to it. The current way of designing the pipeline figure is no different from NuiScene.**
>
> **A5:** We have added the FRC and SGG to Figure 1 in the revison.
>
> **W6: Figure 2 needs a more obvious Z-order direction to be clearer.**
>
> **A6:** We have added more clear the Z-order direction to Figure 2 in the revison.
>
> **W7: Some work on controllable scene generation is missing in L93-94, e.g., CommonScenes, EchoScene, etc.**
>
> **A7:** We have added CommonScenes、EchoScene and InstrucLayout to the related work in the revision.

---

> > ### Comment · Reviewer_1T4i · 2025-11-23
> > **Where is the revison?**
> >
> > Hi, I've just checked your manuscript, but I haven't seen your revision yet. Please upload your updated manuscript BEFORE replying to reviewers if you are saying there is a revision.

---

> > > ### Author Response · Authors · 2025-11-23
> > >
> > > Thanks for your kind remind. We have just corrected the response and updated the revised manuscript.

---

> > ### Comment · Reviewer_1T4i · 2025-11-23
> > **About Lts3d**
> >
> > Hi, thanks for adding LT3SD. The results look indeed weird. Even if you change scenes, the results look similar. I encourage the authors to fix it. However, I do not think the issues come from the indoor/outdoor differences. Even though the heights are not very different from each other in the indoor scenes, the results should be blurred rather than being nonsense. I strongly suggest that you remove Figure 12 from the current revision and reattach it after you have fixed the problem. Otherwise, it is not fair to the previous method.

---

> > ### Comment · Reviewer_1T4i · 2025-11-23
> > **Answers to the questions?**
> >
> > Hi, I would like to ask where the answers to my questions are.  Are you going to answer them?

---

> > > ### Author Response · Authors · 2025-11-24
> > >
> > > We sincerely apologize for the confusion. Although we had prepared detailed responses to your specific questions (Q1--Q3), they were inadvertently excluded from the previous text box due to a pasting error during submission. We provide the full responses immediately below.
> > >
> > > **Q1:Why Only Top and Left Neighbors? Why Not Include Diagonal Context?**
> > >
> > > **A8:** We condition exclusively on Top and Left neighbors for topological and causal reasons.
> > > - **Geometrically**, coherence is defined by shared interfaces (edges), not point contacts (corners). Chunk $(i, j)$ shares physical boundaries only with $(i-1, j)$ and $(i, j-1)$; the diagonal neighbor $(i-1, j-1)$ touches solely at a corner, offering negligible signal for edge continuity.
> > > - **Causally**, given our Z-order traversal, these are the only adjacent chunks fully resolved at inference time. Empirically, ablations (Table 7 in our revision) confirm that regularization on these direct neighbors is sufficient to prevent boundary artifacts. Adding diagonal context increased computational overhead without yielding measurable gains in consistency.
> > >
> > > **Q2: Why Z-order Causal AR for Global, Rather Than Masked AR?**
> > >
> > > **A9:** The choice of Causal AR is dictated by the unbounded nature of the task. Global layouts require an explicit sequential prior to accumulate coherence over arbitrary distances. Masked models, lacking directional dependency, fail to maintain this consistency. Empirical results (Sec. 4.4.2) validate this: without the causal chain, the "Pure Bidirectional MAR" baseline degenerates into repetitive artifacts on large maps. We reserve Masked AR for the local stage, where modeling the joint distribution of unordered tokens is geometrically superior to forcing a sequence. We appreciate this suggestion. Due to resource constraints, we cannot complete a full ablation of a "hierarchical masked AR" during the rebuttal period. However, we commit to including this experiment in the final version to provide a more precise comparison.
> > >
> > > **Q3:It would be better if the authors could demonstrate some texture baking for their examples.**
> > >
> > > **A10:** We agree that texture visualization is a valuable addition. While our primary contribution is high-fidelity geometry synthesis, we treat texturing as an orthogonal task compatible with our standard mesh outputs. We are currently integrating off-the-shelf pipelines (e.g., diffusion baking). However, due to the limited timeline, this adaptation is not yet complete. We will include these results in the final revision.

---

> > > > ### Author Response · Authors · 2025-11-28
> > > > **Update on Baseline Results**
> > > >
> > > > - **XCube:**
> > > >     - **Setup:** We conduct the evaluation at a fixed 8 x 8 chunk scale, strictly following the requirement of XCube for resolution consistency between training and inference phases. **Initial edge-prioritized sampling caused geometric holes. We retrained XCube using dense uniform sampling to ensure sufficient input information.**
> > > >     - **Results:** As shown in Fig. 12, the retrained XCube produces reasonable structures. However, its global embedding VAE formulation struggles to preserve fine details over the large spatial extent of an $8\times8$ scene compared to our chunk-based approach. Additionally, the strict alignment between training and inference resolutions limits its flexibility for large-scale extrapolation.
> > > > - **LT3SD:**
> > > >     - **Setup:** Following the original implementation of LT3SD, we preprocess our 13 outdoor mesh samples and train the LT3SD model accordingly. The voxel size is set to 1.8 meters, with the maximum height of each mesh constrained to under 256 resolution. Apart from the dataset adaptation, we maintain nearly identical training and inference configurations as the original codebase. The LT3SD VAE is trained for 100K iterations, while the diffusion model undergoes 800K iterations of training.
> > > >     - **Results:** We extend our gratitude to the LT3SD authors for their valuable suggestions that facilitated the reproduction of normal results as shown in Fig.12. Through visual inspection of the results, we observe that the scene geometry shows noticeable degradation compared to our ARGOS. Architectural structures and vegetation appear blurred with significant floating artifacts, while the spatial layout demonstrates limited coherence. We attribute the geometry quality decline primarily to resolution constraints. Similar to NuiScene, LT3SD employs voxelization of entire scenes followed by online chunking, which inherently restricts chunk shapes due to voxel conversion parameters. Concerning the spatial incoherence, we hypothesize this stems from the same fundamental limitation as in NuiScene: the outpainting process lacks comprehensive global guidance, resulting in locally optimal but globally suboptimal arrangements.
> > > >
> > > > We appreciate the suggestion to include these baselines. Adapting our data to XCube and LT3SD for a rigorous comparison required additional time. We thank the reviewers for their patience.

---

> > > > ### Author Response · Authors · 2025-12-02
> > > > **Update texture baker results**
> > > >
> > > > For the consider of texture baker, we add two texture baker results generated from ARGOS as shown in Figure 14 of the revised manuscript.  For each untextured mesh, we normalize each scene to $[-1,1]$ and sample 1.6K camera poses in blender platform to build the camera trajectory. Each scene is optimized with H100 for about 12 hours.

---

> ### Author Response · Authors · 2025-11-24
> **About Figure 12**
>
> Thank you for pointing this out. We acknowledge that the current results in Figure 12 are preliminary and do not accurately represent the performance of LT3SD. We are actively working to fix this issue and will update Figure 12 with corrected results in the next revision. To ensure fairness to prior work, we have removed the LT3SD qualitative results from Figure 12 in the current revision. We appreciate your patience and understanding.

---

### Official Review · Reviewer_uf7T · 2025-11-01

**Soundness:** 3
**Presentation:** 3
**Contribution:** 3
**Rating:** 4
**Confidence:** 4

**Summary:**

The authors tackles the challenge of large scale 3d scene generation for outdoor. It developes on top of one prior work, NuiScene, with improvements on (1) input preprocessing procedure in terms of how to densely sample points on chunk mesh, (2) hierarchical generation in terms of global chunk embedding first followed with NuiScene pipeline), and (3) additional text related spatial control. Experiment-wise, they showcase their coarse-to-fine generation pipeline is helpful in the 3Dvector2set chunk features, and they showcase they can do spatial control over text prompt with qualitative results. Finally, they ablate over different generation  paradigm to showcase their procedure can bring the most fidelity 3d scene results.

**Strengths:**

1. One of the key message of this paper is hirerachical generation (coarse-to-fine) is helpful to bring global consistency to large-scale 3d scene generation. It verifies on top of NuiScene overall framework. It demonstrates sufficiently with quantitaive mesaurements (KPD, FPD) in Table 3. The larger, the more significant the difference is. I think this part is clear

2. The proposed contributions are mostly backuped up by qualitative and quantitative evidences, e.g., Coherency-Aware Regularization and Direct Preference Optimization.

3. It is clear that their VAE reconstruction with denser and focused sampling are effective from both qualitative results (Fig. 3 and Table 2)

4. Experiments protocol are detailed discussed with results detailed benchmarked.

**Weaknesses:**

1. Visuals and method descriptions are not clear in a few number of places.
-- Between Line 146 and 148, "To address the limitation, we propose a novel pipeline that directly samples the chunk mesh from the
scene mesh." It brings up some novel pipeline, but it does not describe what exactly it is, and which part is the one the authors refer to as novel pipeline?

-- In Figure 6, what does the highlighted region suppose to indicate? Is it bad or good? It lacks of connection between visuals and captions.

-- In Table 5 and Line 418, I believe it should be FRC instead of RFC.

-- What does Table 1 convey? Similarly, what is take-away from Table 4? The caption seems to be not helpful on analyzing its contents.


2. For the global chunk embeddings generation, it still follows an iterative autoregressive process, i.e., progressive generation paradigm. For z_i^global, in practice, it does not sound feasible to incorporate all previous generated tokens. This part seems not trivial but details are missing.

3. Coarse-to-fine generation has been verified to be helpful in existing 3d scene generation work, e.g., XCube. The novelty might be some concerns for acceptance.


4. It is unclear the performance gap between NuiScene is mainly due to better VAE reconstruction or latter coarse-to-fine generation. If we compare the visual results between Nuiscene (16x160 and theirs, the qualitative results are hard to tell which is better. If the gap mostly lies in data sampling when preparing the features, it is a bit unclear how necessary we need to do the global chunk embedding first, and then detailed chunk + diffusion later.

5. The qualitative results are not very convincing. All qualitative results are only given 1 example, which makes the conclusion questionable for generalization. For example, the control of contents are not very clear with only one text prompt example.

6. The comparisons with key 3d scene gen baselines are missing. For example, Earthcraft (https://arxiv.org/abs/2507.16535). XCube (https://arxiv.org/pdf/2312.03806). LT3SD (https://openaccess.thecvf.com/content/CVPR2025/papers/Meng_LT3SD_Latent_Trees_for_3D_Scene_Diffusion_CVPR_2025_paper.pdf).

**Questions:**

1. It would be nice to have additional comparison on the same feature derived from the same process of input mesh. And then feed those to Nuiscene and the authors proposed generation pipeline. I think it could be helpful to reveal how much the hierarhical process really help compared with their key baseline.

2. It would also be great to have additional comparison over other SOTA 3d scene baselines.

---

> ### Author Response · Authors · 2025-11-23
>
> We thank the reviewer for their constructive feedback. We address the concerns regarding method clarity, global AR feasibility, and baseline comparisons below.
>
> **W1: Visuals and method descriptions are not clear.**
>
> **A1:** We clarify the specific details regarding the data pipeline and figures:
>
> - **The Novel Pipeline.** The "novel pipeline" (Sec. 3.1in main paper) refers to our ​mesh-based occupancy sampling strategy. Unlike NuiScene, which extracts chunks from scene-level occupancy, we sample occupancy directly from cropped mesh segments (using the NuiScene sampling mask, as illustrated in Figure 7 of the Appendix). This approach decouples data generation from scene-level memory constraints, enabling supervision at higher resolutions ($256^3$) and significantly improving VAE reconstruction accuracy.
> - **Figure 6 Annotations.** The highlighted regions in Figure 6 explicitly denote geometric artifacts and inconsistencies, contrasting with results of our final ARGOS.
> - **Notation Correction.** We have corrected the notation in Table 5 and Line 418 to accurately reflect "FRC" (Flexible Resolution Control) in the revision.
> - **Take-aways from Tables 1 & 4.** Table 1 illustrates the diversity of our multi-resolution training data. Table 4 quantitatively validates that this diversity translates to superior generalization: ARGOS achieves higher CLIP and Uni3D scores, statistically demonstrating robust spatial control across varying scales. We have augmented the captions in the revision to explicitly support this analysis.
>
> **W2: Global AR Feasibility**
>
> **A2:** We clarify the computational feasibility of our global autoregressive generation.
>
> Our global generation follows a standard autoregressive process. While Eq. 4 in our main paper formulates the dependency on all previous tokens (\$z^{global}\_{<i}\$), the effective context length is determined by the training resolution.
>
> - **Context Window:** Our model is trained on scenes up to 16 x16 chunks. With the VAE compressing each chunk to $l=16\$ tokens, the maximum context window is 4096 tokens. This is well within the capacity of modern Transformer architectures.
> - **Inference Strategy:** For unbounded generation exceeding the training size (e.g., the 32 x 32 experiments), we employ a sliding window approach, attending to the most recent 4096 tokens.
>
> - **Enabler:** This feasibility is directly enabled by our ​the high compression ratio of our VAE. Autoregressively modeling raw $256^3\$ grids  is computationally prohibitive. Our VAE compresses a chunk into just 16 tokens while maintaining geometric fidelity, making global context modeling tractable.
>
> **W3: Novelty of Hierarchical Framework vs. XCube**
>
> **A3:** We distinguish our framework from XCube in terms of ​concept​, ​necessity and ​implementation​.
>
> - **Concept: Structural vs. Resolutional.** XCube utilizes a resolutional hierarchy to solve a resolution problem. In contrast, ARGOS employs a structural hierarchy to solve a ​long-range coherence problem​. We decouple generation into two distinct tasks: implicit global planning (Causal AR) and local detail geometric generation (Masked AR).
> - **Necessity.** Our ablations (Sec. 4.4.2 in main paper) demonstrate that this structural decomposition is essential for unbounded generation. A global-only model ("Block-wise Causal AR") yields consistently coarse outputs, while a local-only model ("Bidirectional MAR") suffers from repetitive patterns and structural collapse at large scales.
>
> * **Implementation: Implicit vs. Explicit.** ARGOS uses a Single-VAE framework where the global layout $z^{global}\$ is an implicit representation learned purely by the AR model. XCube relies on a Multi-VAE setup with explicit ground-truth latents at each resolution level. Our design is specifically optimized for unbounded, streaming generation, whereas XCube targets bounded, high-resolution volume super-resolution.

---

> ### Author Response · Authors · 2025-11-23
>
> **W4: It is unclear the performance gap between NuiScene is mainly due to better VAE reconstruction or latter coarse-to-fine generation.**
>
> **A4:** We clarify the contribution sources through additional experiments.
>
> - **Existing Evidence.** Our main paper provides initial separation:
>
>   - Table 2 shows ARGOS VAE improves reconstruction quality
>   - Table 3 compares generation frameworks using the same improved VAE, where ARGOS outperforms retrained NuiScene (16×16), especially at large scales
> - **Additional Experiment.** For stricter comparison, we conduct a controlled experiment. Specifically, we train the ARGOS AR generator on the original frozen NuiScene VAE at its native resolution\~(\$50, h\_{vox}, 50\$), keeping all other settings unchanged.
> **Table: Quantitative Comparison using Frozen NuiScene VAE (Single-Scene)**
>
> | Method | 2×2(↓) | 8×8(↓) | 32×32(↓) |
> |--------|-----|-----|-------|
> |        | FPD/KPD* | FPD/KPD* | FPD/KPD* |
> | NuiScene original| **0.049/0.11** | 0.37/0.48 | 1.26/1.33 |
> | **ARGOS + NuiScene VAE** | 0.06/0.22 | **0.33/0.41** | **0.93/0.98** |
>
> Above table presents the results on single-scene data. Despite identical reconstruction baselines, ARGOS outperforms the NuiScene baseline significantly at large scales (32 x 32). While performance is comparable at small scales (2 x2), ARGOS exhibits superior scalability as scene complexity increases. Qualitative comparisons( refer to Figure 6 in revision manuscript) (13-scene data) further demonstrate that ARGOS generates coherent layouts with reduced repetitive patterns. These results confirm that our hierarchical design is the primary contributor to long-range coherence, independent of VAE fidelity.
>
> **W5: The qualitative results are not very convincing.**
>
> **A5:** We have enriched the manuscript with expanded qualitative comparisons, refer to Sec A.6:
>
> - **Unconditional Generation:** We provide new multi-scene visualizations contrasting ARGOS against NuiScene. These results explicitly illustrate the long-range coherence of ARGOS, effectively resolving the fragmented transitions and boundary artifacts observed in the baseline.
> - **Conditional Generation:**  We present generation results for 4 diverse prompts (3scenes per prompt). These samples verify that ARGOS achieves geometric diversity, while maintaining rigorous adherence to the prescribed spatial constraints.
>
> **W6: The comparisons with key 3d scene gen baselines are missing.**
>
> **A6:** Earthcraft  is not open-source and we have conducted a preliminary comparison with XCube and LT3SD.
>
> - **XCube:**
>   - **Setup:** We conduct the evaluation at a fixed 8 x 8 chunk scale, strictly following the requirement of XCube for resolution consistency between training and inference phases.
>   - **Results:** We present results derived from the official XCube preprocessing, as shown in Figure 12 of revision manuscript. These outputs exhibit geometric holes on ground planes, empirically indicating that XCube is inherently optimized for bounded objects rather than continuous outdoor surfaces. Specifically, the sampling routine prioritizes geometric edges, resulting in insufficient density on large planar regions . Consequently, this sparsity directly induces the observed artifacts. To ensure a robust baseline, we are currently retraining XCube with adapted dense uniform sampling and will report the optimized results.
> - **LT3SD:**
>   - **Setup:** LT3SD is mainly conducted in the indoor dataset 3DFront and we follow its codebase to preprocess our ourdoor mesh data and train the LT3SD model. We consider mesh chunk with 8x8 chunk size as the scene shape and sample 5K mesh chunk for LT3SD training. Each mesh chunk is processed with resolution 512³ .
>   - **Results:** We have obtained a coarse result, but the result seems strange. We are currently investigating the issue and are unsure whether it stems from data processing problems or training and inference configuration issues. The Lt3sd code was primarily designed for indoor scene generation, which typically involves limited height variations, whereas outdoor scenes require handling diverse elevation changes and would likely need an additional adaptation process. In any case, we are striving to achieve normal results with Lt3sd on our dataset to enable a comparative analysis.
>
> **Q1: Additional comparison on the same feature derived from the same process of input mesh.**
>
> **A7:** This question is the same as part of the **W4** and please refer to **A4**.
>
> **Q2: Additional comparison over other SOTA 3d scene baselines.**
>
> **A8:** Please refer to **A6**.

---

> ### Comment · Reviewer_uf7T · 2025-11-26
>
> Thanks for the detailed feedback! The answers help clear some of my confusions on its merits, e.g., A4. Yet, there are still concerns for me to accept this paper.
> - I found the additional qualitative and (then quantitative) related to XCube weak. For Fig.12, your XCube qualitative results do not match what was seen from their original paper. In the original scene they have outdoor cases as well, and produce much better results than the result here used for comparison.
> - I echoed with Reviewer aDSy, I cannot explain why Fig. 7 marked region is bad. similarly, for the additional qualitative examples, it is hard to argue why Nuiscene 16x16 is worse.
>
> I think the current deliverable is good for ablation but not strong in terms of comparison experiments

---

> > ### Author Response · Authors · 2025-11-28
> > **Response to Visual Ablation and Comparisons**
> >
> > - **Visual Ablation Analysis**
> > Since the failure of Block-wise Causal AR is evident, we focus on **Bidirectional MAR**, **w/o DPO**, and **w/o CA (Coherency-Aware) loss**. Beyond Fig.7 in the main paper, we provide additional $8\times8$ results (5 samples per model) in Fig. 13 of the revised manuscript to visualize stability.
> >     - **Bidirectional MAR:** Lacks explicit global guidance, preventing long-range consistency. Fig.7(b) highlights discontinuous element repetition (red box, bottom-left) and structural collapse in large scenes (top-right). Fig.13(a) confirms this, showing cluttered layouts and large-scale corruption (Row 3).
> >     - **w/o DPO Loss:** Suffers from mode collapse. The model favors simplistic patterns, such as repetitive flat ground or substantial empty space. This is marked in Fig.7(c) and evident in Fig.13(b), which exhibits a larger proportion of empty regions.
> >     - **w/o CA Loss:** Addresses relatively subtle consistency issues, as reflected in the quantitative metrics of Table 7. Its absence introduces interruptions at chunk boundaries, accompanied by slight incoherent element repetition. Fig.7(d) shows incoherent repetition of "bridge" elements. Fig.13(c) illustrates similar artifacts: Row 1 highlights element repetition and subsequent rows show unreasonable structural interruptions.
> > - **Visual Comparison with Baselines**
> >     - **NuiScene ($16\times16$):** Fig.10 shows NuiScene suffers from frequent element repetition (e.g., containers in Row 2, bridges in Row 3) and structural collapse (e.g., buildings in Row 3). Fig. 12 further reveals global layout incoherence, such as the disconnected combinations in Row 3. In contrast, ARGOS maintains global coherence with preserved local details.
> >     - **XCube:**
> >         - **Setup:** We conduct the evaluation at a fixed 8 x 8 chunk scale, strictly following the requirement of XCube for resolution consistency between training and inference phases. **Initial edge-prioritized sampling caused geometric holes. We retrained XCube using dense uniform sampling to ensure sufficient input information.**
> >         - **Results:** As shown in Fig. 12, the retrained XCube produces reasonable structures. However, its global embedding VAE formulation struggles to preserve fine details over the large spatial extent of an $8\times8$ scene compared to our chunk-based approach. Additionally, the strict alignment between training and inference resolutions limits its flexibility for large-scale extrapolation.
> >     - **LT3SD:**
> >         - **Setup:** Following the original implementation of LT3SD, we preprocess our 13 outdoor mesh samples and train the LT3SD model accordingly. The voxel size is set to 1.8 meters, with the maximum height of each mesh constrained to under 256 resolution. Apart from the dataset adaptation, we maintain nearly identical training and inference configurations as the original codebase. The LT3SD VAE is trained for 100K iterations, while the diffusion model undergoes 800K iterations of training.
> >         - **Results:** We extend our gratitude to the LT3SD authors for their valuable suggestions that facilitated the reproduction of normal results as shown in Fig.12. Through visual inspection of the results, we observe that the scene geometry shows noticeable degradation compared to our ARGOS. Architectural structures and vegetation appear blurred with significant floating artifacts, while the spatial layout demonstrates limited coherence. We attribute the geometry quality decline primarily to resolution constraints. Similar to NuiScene, LT3SD employs voxelization of entire scenes followed by online chunking, which inherently restricts chunk shapes due to voxel conversion parameters. Concerning the spatial incoherence, we hypothesize this stems from the same fundamental limitation as in NuiScene: the outpainting process lacks comprehensive global guidance, resulting in locally optimal but globally suboptimal arrangements.
> >
> > We appreciate the suggestion to include these baselines. Adapting our data to XCube and LT3SD for a rigorous comparison required additional time. We thank the reviewers for their patience.

---

### Author Response · Authors · 2025-11-23

We thank the reviewers for their constructive feedback. Below we summarize the common concerns raised by the reviewers.

**Q1: Novelty**

**A1:** We explicitly define our novelty across three dimensions, clarifying our technical contributions

- **Hybrid AR: Structural Hierarchy.** Our framework addresses the tension in unbounded generation: maintaining global consistency while preserving local detail. We decouple this into two stages: a Causal AR establishes a globally coherent implicit layout, followed by a Masked AR that synthesizes detailed geometry conditioned on this layout. This Single-VAE, implicit-planning design fundamentally distinguishes ARGOS from approaches relying on explicit multi-resolution ground-truth latents. To isolate this contribution, we add a controlled experiment training ARGOS with the frozen NuiScene VAE. Results show an FPD improvement from 1.26 to 0.93 at the 32 x 32 scale, independent of reconstruction fidelity (refer to Q3 / Revision Sec. 4.3). This confirms that our hierarchical design is the primary driver of long-range coherence.
- **VAE: High-Fidelity Supervision Pipeline.** While sharing the core architecture of 3DShape2VectorSet, our contribution lies in the supervision pipeline. We resolve the resolution bottleneck of conventional mesh2sdf approaches (typically limited to 50³) by sampling occupancy from cropped mesh segments rather than scene-level volumes (refer to Revision Sec. A.1). This strategy bypasses previous memory constraints, enabling supervision at 256³ resolution. Tables 2 and 5 of the main paper verify that this high-resolution supervision significantly enhances reconstruction quality, offering a scalable solution for high-fidelity chunk representation.
- **Conditional Generation: Structured Spatial Control.**  We mitigate the ambiguity of text-to-3D generation via an automated pipeline that extracts topological constraints. This bridges flexible natural language with rigid geometric definitions, yielding high diversity and strict adherence to spatial arrangements. We acknowledge, however, that natural language is inherently under-constrained for precise metric adherence. We envision future work addressing this trade-off by integrating hybrid control signals—fusing the semantic breadth of text with the determinism of bounding boxes. We have expanded the discussion of these modality-specific limitations in the revision manuscript.

**Q2: Motivation**

**A2:** We articulate the necessity of our hierarchical architecture compared to Diffusion and Flat AR baselines:

- **vs. Diffusion:** Diffusion models typically rely on local inpainting windows. They inherently lack an explicit state to propagate spatial history beyond the immediate context, leading to semantic drift and visible boundary artifacts in unbounded generation. In contrast, ARGOS utilizes Causal AR to condition on the ​compressed global history, enforcing logical consistency across the entire scene sequence.
- **vs. Flat AR :** Modeling raw 3D tokens in a single stream forces a trade-off: simultaneously resolving large-scale structure and high-frequency detail is optimizationally intractable. Our ablations (refer to Table 6 in the main paper) confirm this: "Block-wise Causal AR" yields overly coarse geometry, whereas "Pure Bidirectional MAR" collapses into repetitive patterns at scale. Our decomposition is essential, assigning global planning and local synthesis to the modeling paradigms best suited for each.

**Q3: Architectural Decomposition: Hybrid AR vs. VAE**

**A3:** To decouple the contribution of our hierarchical AR framework from VAE improvements, we conduct a controlled experiment. Specifically, we train the ARGOS AR generator on the original frozen NuiScene VAE at its native resolution~($50, h_{vox}, 50$), keeping all other settings unchanged.

**Table: Quantitative Comparison using Frozen NuiScene VAE (Single-Scene)**

| Method | 2×2(↓) | 8×8(↓) | 32×32(↓) |
|--------|-----|-----|-------|
|        | FPD/KPD* | FPD/KPD* | FPD/KPD* |
| NuiScene original| **0.049/0.11** | 0.37/0.48 | 1.26/1.33 |
| **ARGOS + NuiScene VAE** | 0.06/0.22 | **0.33/0.41** | **0.93/0.98** |

Above table presents the results on single-scene data. Despite identical reconstruction baselines, ARGOS outperforms the NuiScene baseline significantly at large scales (32 x 32). While performance is comparable at small scales (2 x2), ARGOS exhibits superior scalability as scene complexity increases. Qualitative comparisons( refer to Figure 6 in revision manuscript) (13-scene data) further demonstrate that ARGOS generates coherent layouts with reduced repetitive patterns. These results confirm that our hierarchical design is the primary contributor to long-range coherence, independent of VAE fidelity.

---

> ### Author Response · Authors · 2025-11-23
>
> **Q4: New Baseline (XCube & LT3SD)**
>
> **A4:** We have conducted a preliminary comparison with XCube and LT3SD.
>
> - **XCube:**
>   - **Setup:** We conduct the evaluation at a fixed 8 x 8 chunk scale, strictly following the requirement of XCube for resolution consistency between training and inference phases.
>   - **Results:** We present results derived from the official XCube preprocessing, as shown in Figure 12 of revision manuscript. These outputs exhibit geometric holes on ground planes, empirically indicating that XCube is inherently optimized for bounded objects rather than continuous outdoor surfaces. Specifically, the sampling routine prioritizes geometric edges, resulting in insufficient density on large planar regions . Consequently, this sparsity directly induces the observed artifacts. To ensure a robust baseline, we are currently retraining XCube with adapted dense uniform sampling and will report the optimized results.
> - **LT3SD:**
>   - **Setup:** LT3SD is mainly conducted in the indoor dataset 3DFront and we follow its codebase to preprocess our ourdoor mesh data and train the LT3SD model. We consider mesh chunk with 8x8 chunk size as the scene shape and sample 5K mesh chunk for LT3SD training. Each mesh chunk is processed with resolution 512³ .
>   - **Results:** We have obtained a coarse result, but the result seems strange. We are currently investigating the issue and are unsure whether it stems from data processing problems or training and inference configuration issues. The Lt3sd code was primarily designed for indoor scene generation, which typically involves limited height variations, whereas outdoor scenes require handling diverse elevation changes and would likely need an additional adaptation process. In any case, we are striving to achieve normal results with Lt3sd on our dataset to enable a comparative analysis.
>
> **Q5: Enhanced Visual Evidence & Diversity**
>
> **A5:** Earthcraft  is not open-source and we have enriched the manuscript with expanded qualitative comparisons, refer to Sec A.6:
>
> - **Unconditional Generation:** We provide new multi-scene visualizations contrasting ARGOS against NuiScene. These results explicitly illustrate the long-range coherence of ARGOS, effectively resolving the fragmented transitions and boundary artifacts observed in the baseline.
> - **Conditional Generation:** We present generation results for 4 diverse prompts (3scenes per prompt). These samples verify that ARGOS achieves ​geometric diversity, while maintaining rigorous adherence to the prescribed spatial constraints.
>
> Overall, we have incorporated the following experiments and visualizations into the revised manuscript to address reviewer feedback:
>
> * **Table 4:** Quantitative comparison using frozen NuiScene VAE.
> * **Figure 6:** Qualitative comparison of long-range coherence (13-scene data) using the frozen VAE.
> * **Figure 8:** Visualization of raw mesh for large-scale scenes.
> * **Figure 10:** Extended qualitative results for unconditional generation.
> * **Figure 11:** Extended samples for text-conditioned generation .

---

### Author Response · Authors · 2025-11-28
**Update on Visual Ablation and Comparisons with Baselines**

Based on the reviewer's follow-up feedback and our commitment in the Overall Response, we have updated the visual ablation and baseline comparison sections below. We have also submitted a revised PDF, incorporating an updated **Figure 12** and a newly added **Figure 13** for visual ablation.

- **Visual Ablation Analysis**
  Since the failure of Block-wise Causal AR is evident, we focus on **Bidirectional MAR**, **w/o DPO**, and **w/o CA (Coherency-Aware) loss**. Beyond Fig.7 in the main paper, we provide additional $8\times8$ results (5 samples per model) in Fig. 13 of the revised manuscript to visualize stability.
     - **Bidirectional MAR:** Lacks explicit global guidance, preventing long-range consistency. Fig.7(b) highlights discontinuous element repetition (red box, bottom-left) and structural collapse in large scenes (top-right). Fig.13(a) confirms this, showing cluttered layouts and large-scale corruption (Row 3).
    - **w/o DPO Loss:** Suffers from mode collapse. The model favors simplistic patterns, such as repetitive flat ground or substantial empty space. This is marked in Fig.7(c) and evident in Fig.13(b), which exhibits a larger proportion of empty regions.
    - **w/o CA Loss:** Addresses relatively subtle consistency issues, as reflected in the quantitative metrics of Table 7. Its absence introduces interruptions at chunk boundaries, accompanied by slight incoherent element repetition. Fig.7(d) shows incoherent repetition of "bridge" elements. Fig.13(c) illustrates similar artifacts: Row 1 highlights element repetition and subsequent rows show unreasonable structural interruptions.

- **Visual Comparison with Baselines**
    - **NuiScene ($16\times16$):** Fig.10 shows NuiScene suffers from frequent element repetition (e.g., containers in Row 2, bridges in Row 3) and structural collapse (e.g., buildings in Row 3). Fig. 12 further reveals global layout incoherence, such as the disconnected combinations in Row 3. In contrast, ARGOS maintains global coherence with preserved local details.
    - **XCube:**
        - **Setup:** We conduct the evaluation at a fixed 8 x 8 chunk scale, strictly following the requirement of XCube for resolution consistency between training and inference phases. **Initial edge-prioritized sampling caused geometric holes. We retrained XCube using dense uniform sampling to ensure sufficient input information.**
        - **Results:** As shown in Fig. 12, the retrained XCube produces reasonable structures. However, its global embedding VAE formulation struggles to preserve fine details over the large spatial extent of an $8\times8$ scene compared to our chunk-based approach. Additionally, the strict alignment between training and inference resolutions limits its flexibility for large-scale extrapolation.

    - **LT3SD:**
        - **Setup:** Following the original implementation of LT3SD, we preprocess our 13 outdoor mesh samples and train the LT3SD model accordingly. The voxel size is set to 1.8 meters, with the maximum height of each mesh constrained to under 256 resolution. Apart from the dataset adaptation, we maintain nearly identical training and inference configurations as the original codebase. The LT3SD VAE is trained for 100K iterations, while the diffusion model undergoes 800K iterations of training.
        - **Results:** We extend our gratitude to the LT3SD authors for their valuable suggestions that facilitated the reproduction of normal results as shown in Fig.12. Through visual inspection of the results, we observe that the scene geometry shows noticeable degradation compared to our ARGOS. Architectural structures and vegetation appear blurred with significant floating artifacts, while the spatial layout demonstrates limited coherence. We attribute the geometry quality decline primarily to resolution constraints. Similar to NuiScene, LT3SD employs voxelization of entire scenes followed by online chunking, which inherently restricts chunk shapes due to voxel conversion parameters. Concerning the spatial incoherence, we hypothesize this stems from the same fundamental limitation as in NuiScene: the outpainting process lacks comprehensive global guidance, resulting in locally optimal but globally suboptimal arrangements.

We appreciate the suggestion to include these baselines. Adapting our data to XCube and LT3SD for a rigorous comparison required additional time. We thank the reviewers for their patience.

---

### Meta-Review · Area_Chair_3LuJ · 2025-12-19

**Summary:**

The main reason for my decision is motivated by the critical deficiencies in the experimental validation that persisted after the rebuttal. The comparisons with state-of-the-art baselines (XCube and LT3SD) introduced during the discussion were widely criticized; reviewers characterized the provided baseline results as "broken" or visually nonsensical, undermining the credibility of the claimed improvements and raising concerns about evaluation practices. Additionally, the qualitative evidence remained unconvincing, with multiple reviewers stating they could not perceive the specific artifacts or boundary inconsistencies in the control methods that the authors' visual ablations sought to demonstrate. These experimental flaws, combined with the structural limitation of training on a small, curated 13-scene dataset, fail to sufficiently demonstrate the method's generalization capabilities or its superiority over existing approaches.

**Reviewer Concerns:**

The rebuttal clarified the architectural motivation, specifically why an autoregressive approach is necessary for maintaining global consistency in unbounded generation compared to diffusion-based inpainting. The authors also provided the requested inference latency and memory statistics. The ambiguity regarding the source of performance gains was effectively addressed through a new controlled experiment.

However, significant issues regarding the comparative evaluation remain unresolved. The baseline comparisons added during the rebuttal (XCube and LT3SD) were heavily criticized, with reviewers characterizing the results as "broken" or visually nonsensical, leading to concerns about unfair evaluation practices. Additionally, the visual ablations failed to convince multiple reviewers, who stated they could not perceive the specific boundary artifacts or inconsistencies the authors claimed were present in the baseline methods. Finally, the reliance on a small, curated 13-scene dataset remains a structural weakness, leaving the model's generalization capabilities unproven against broader datasets used in prior work.

**Reviewer Scores:**

Reviewer uf7T: In their post-rebuttal comments, they explicitly stated that "there are still concerns for me to accept," citing the XCube comparisons as "weak" and inconsistent with the original paper's quality.

Reviewer 1T4i: Despite their initial positive outlook, their reaction to the rebuttal data was severe; they described the new LT3SD baseline results as "nonsense" and "broken," criticizing the authors for unfair comparisons.

Reviewer hjzj:  Their primary concerns centered on the small dataset size and generalization, which the authors defended by framing it as a controlled comparison against NuiScene.

Reviewer aDSy: Their post-rebuttal engagement indicated continued confusion regarding the visual ablations, as they explicitly stated they could not perceive the boundary inconsistencies the authors claimed to solve.

---

### Decision · Program_Chairs · 2026-01-26

Reject